# Marine Animal Co-Products—How Improving Their Use as Rich Sources of Health-Promoting Lipids Can Foster Sustainability

**DOI:** 10.3390/md22020073

**Published:** 2024-01-30

**Authors:** João Pedro Monteiro, M. Rosário Domingues, Ricardo Calado

**Affiliations:** 1Centro de Espetrometria de Massa, LAQV-REQUIMTE, Departamento de Química, Universidade de Aveiro, Campus Universitário de Santiago, 3810-193 Aveiro, Portugal; 2CESAM, Departamento de Química, Universidade de Aveiro, Campus Universitário de Santiago, 3810-193 Aveiro, Portugal; 3ECOMARE, CESAM, Departamento de Biologia, Universidade de Aveiro, Campus Universitário de Santiago, 3810-193 Aveiro, Portugal

**Keywords:** bioactive lipids, lipidomics, marine animal co-products, omega-3 PUFAs, seafood industry, sustainable practices

## Abstract

Marine lipids are recognized for their-health promoting features, mainly for being the primary sources of omega-3 fatty acids, and are therefore critical for human nutrition in an age when the global supply for these nutrients is experiencing an unprecedent pressure due to an ever-increasing demand. The seafood industry originates a considerable yield of co-products worldwide that, while already explored for other purposes, remain mostly undervalued as sustainable sources of healthy lipids, often being explored for low-value oil production. These co-products are especially appealing as lipid sources since, besides the well-known nutritional upside of marine animal fat, which is particularly rich in omega-3 polyunsaturated fatty acids, they also have interesting bioactive properties, which may garner them further interest, not only as food, but also for other high-end applications. Besides the added value that these co-products may represent as valuable lipid sources, there is also the obvious ecological upside of reducing seafood industry waste. In this sense, repurposing these bioresources will contribute to a more sustainable use of marine animal food, reducing the strain on already heavily depleted seafood stocks. Therefore, untapping the potential of marine animal co-products as valuable lipid sources aligns with both health and environmental goals by guaranteeing additional sources of healthy lipids and promoting more eco-conscious practices.

## 1. Introduction

The growing demand for sustainable and health-promoting food sources has prompted a reevaluation of underutilized resources within the seafood industry. It is estimated that only 50 to 60% of the product of marine animal catch end up being used for direct human consumption, and therefore, the production and disposal of co-products raises problems related to industrial logistics, environmental impact, and even human health [1,2,3]. In an era where concerns about overfishing, waste generation and disposal, and environmental sustainability are paramount, the utilization of marine animal co-products emerges as a logical and appealing solution. The rational use of these co-products may present viable and relevant solutions to both nutritional and environmental challenges, and the identification of additional valorization routes for these raw materials may also come with economic upside [4]. In this review, we will explore the expanding field of marine animal co-products and their potential as sustainable sources of healthy and bioactive lipids, providing a holistic perspective linking health and environmental objectives. It aims to represent a preamble for researchers, policymakers, and industry stakeholders interested in exploring the potential of marine animal co-products as valuable healthy lipid sources with economic upsides, while contributing to reduce and minimize the ecological impacts that still persist in the seafood industry.

Marine animal co-products encompass a diverse range of materials generated during seafood processing, covering a variety of species, including fish, crustaceans, and mollusks. They include biological matrixes like fish/crustacean/mollusk heads, bones, skin, shells, and viscera, which are at times discarded or otherwise underutilized. This vast resource pool represents a potential reservoir of lipids, including essential fatty acids, omega-3 fatty acids, and other bioactive lipids waiting to be explored. It is estimated that a third of the total omega-3 PUFA eicosapentaenoic acid (EPA) + docosahexaenoic acid (DHA) originated from global capture and aquaculture operations may still go to waste [5]. Therefore, considering the generally described benefits of marine animal fats, including counteracting manifestations of cardiovascular disease [6,7,8,9] and metabolic syndrome [10,11,12,13] as well as inflammation [13,14,15,16] while enhancing cognitive function [17,18,19], these co-products represent additional resources that may contribute to the mitigation of important widespread conditions promoted by the modern lifestyle. Fat from some marine animal co-products (especially the ones from fish) is already being explored for the production of fish oils, but these often represent low-value end products. Fish oils may be produced from fish co-products (especially viscera) through a number of different processes, including rendering, pressing, microwave-assisted extraction, supercritical fluid extraction, solvent extraction, autolysis, and enzymatic hydrolysis [20]. The fish oil market represented a total of USD 1905.77 million in 2019 and is continually growing [21], with fish oil produced from fish co-products representing 26% of total production in 2016 [22]. However, the exploration of lipid fractions for other purposes than oil production alone, including the retrieval of phospholipid-rich fractions, the prospection for bioactivities, and the identification of bioactive lipids, may pave the way for the development of specialized products targeting specific applications and/or human conditions. This will promote the repurposing of these co-products for higher-end applications, contributing to their valorization in additionally profitable markets. The diversity of marine co-products, as well as the multitude and origin of species and the processing methods involved, can result in variations in lipid composition. Therefore, a more thorough characterization of marine animal co-product lipids will enable researchers and industry stakeholders to better tailor their applications, whether for nutritional enrichment, functional food development, or pharmaceutical purposes.

The global marine co-product market was proposed to represent USD 33.7 million in 2023, with the current projections anticipating this value to almost double by 2033 (USD 64.8 million), with this growth being proposed to be essentially driven by the expansion of the mariculture sector, technical advances in extraction and processing techniques, and the development of novel applications as functional foods and nutraceuticals [23]. The importance of a rational and holistic perspective for the repurposing of marine animal co-products may be essential for the generation of revenue. A study appraising the effectiveness of the Scottish salmon farming system proposed that, although resources are generally well utilized, the co-product value output could be improved by 803%, representing 5.5% more value to the salmon industry [24]. This could be achieved by a strategic management of co-products (heads, frames, trimmings, and belly flaps) focusing on optimizing edible yield and repurposing and directing them specifically for domestic and foreign food markets [24]. In fact, while salmon co-products may not be especially appreciated in some markets, they may be in high demand in others, and this is the case of salmon heads in Vietnam, frames in Eastern Europe, and belly flaps in Japan [1].

Simultaneously, the sustainability aspect of marine animal co-product utilization cannot be understated. The seafood industry faces increasing challenges related to resource depletion and waste generation and disposal. Overfishing/overharvesting represent, obviously, the most overarching problems, with many species at risk of depletion due to unsustainable practices [25,26,27,28]. By repurposing marine animal co-products as valuable lipid sources, this approach contributes to waste reduction, contributes to marine ecosystem conservation, and supports a more responsible and holistic resource management. Furthermore, a strategic use of marine animal co-products could play a role in fostering the circular economy within the seafood industry, a vital step towards reducing the industrial ecological footprint.

## 2. Marine Animal Co-Products

Seafood encompasses a large and varied range of different animal species, including fish (e.g., salmon, tuna, cod, sardine, seabass, seabream), crustaceans (e.g., crabs, shrimp, and lobsters), and bivalve (e.g., mussels, clams, and oysters) and cephalopod mollusks (e.g., squid and octopus). They can arrive to the market either through farming (aquaculture) or being wild-caught (fishery operations), with both production systems being collectively responsible for 178 million tons of aquatic animals in 2020 [29]. In the specific case of fish, aquaculture production (94.7 million metric tons) is expected to have slightly surpassed production by capture fisheries (90.7 million metric tons) in 2023 [30]. Most commonly, consumer interest in seafood products primarily focuses on fillets and whole seafood items, such as shrimp or fish [31]. In many occasions, however, a substantial portion of harvested marine animals, corresponding to less valuable, non-edible, and less marketable parts, is depreciated as co-products [32]. Marine animal co-products may be generated during all the production, processing, distribution, consumption, and disposal stages, with the processing stage being responsible for the majority of unused material produced [33,34]. Normally, seafood processing co-products result from processes that include bleeding, beheading, deshelling, skinning, trimming, gutting, removal of fins and scales, filleting, and washing [35,36] (Figure 1). Within the context of marine animal co-products, it is crucial to understand the intricacies of this resource pool. Each specific co-product may present unique opportunities and challenges for lipid extraction and repurposing, and therefore, a detailed examination of their detailed profiles and characteristics, specifically using more advanced lipidomics methodologies, is mandatory.

Seafood industries have made some efforts to deal with the significant amounts of biomass originated by their activity. Still, the level of biowaste generated is commonly either repurposed for low-value purposes, such as animal feed, plant fertilizers, fish oils and fish meals, or even biodiesel production, or is simply discarded, often being incinerated, resulting in additional energy consumption, costs, and environmental impact [21,37,38,39]. Therefore, the current scenario is far from ideal for both industry players and the general population, representing a missed opportunity from an economic standpoint, while also contributing to environmental issues related to increased waste disposal and processing. Therefore, understanding and repurposing the various types of marine animal co-products and their potential for sustainable resource development is crucial from both the economic and ecological standpoints. Moreover, by recognizing the diversity of marine animal co-products and their local/regional availability, stakeholders can devise resource utilization strategies to minimize waste and maximize value within the seafood industry while guaranteeing foreseeable benefits for local economies.

Taking into account idiosyncratic geographical tendencies regarding both production/capture and consumption trends driven by regional socioeconomic and cultural constraints, differences in the production of co-products, both regarding quantity and type, are to be expected. According to the IFFO—The Marine Ingredients Organization site, Asia is responsible for recycling the majority of raw materials from the seafood industry, accounting for the production of 40.0% of marine ingredients based on co-products, followed by Europe (23.3%) and Latin America (20.8%), with North America producing just 7.8% of global co-products [40]. This is somewhat expected since in 2020, Asian countries distinctly led total fishery production (70%) [41]. Although considerable efforts have been made to more readily utilize marine animal co-products originated from the seafood processing worldwide, their exploration and use may vary considerably regionally [42]. In Asia, seafood value chains effectively accommodate these resources, resulting in little waste, while in Europe, stricter legislation (as response to bovine spongiform encephalopathy and other food threats) has rendered the utilization of these resources more difficult [43,44]. In other, less developed regions, more lenient legislation and the low valorization of co-products make disposal more generalized [42]. A specific case of particularly efficient utilization of marine animal co-products is Norway, which has developed processing facilities able to process over 0.65 million tons of seafood co-products by year, and where the Norwegian Atlantic salmon industry is reported to utilize 90% of its byproducts [45].

### 2.1. Fish

According to the Food and Agriculture Organization of the United Nations (FAO) in the latest “The State of World Fisheries and Aquaculture 2022” report, global marine and diadromous fish production amounted to 77 million tons in 2020 (Figure 2) [29]. Approximately 67 million metric tons of marine/diadromous fish were obtained via capture, while only around 10 million tons were produced in aquaculture. The most consumed fishes in the European Union are tuna (several species, including *Katsuwonus pelamis*, *Thunnus albacares,* and *Thunnus thynnus*), salmon (*Salmo salar*), cod (*Gadus morhua*), and Alaska pollock (*Gadus chalcogrammus*), in this respective order [46]; therefore, these should be the species responsible for generating the greatest amounts of co-products. Moreover, it is estimated that around 70% of the fish captured and produced undergoes processing before entering the market, which implies a considerable production of potential fish waste if this biomass is not duly used for other purposes [47]. These processing steps preceding human consumption, intended to facilitate consumer manipulation, are normally achieved by the removal of uneatable or less savory parts. This processing originates co-products in the form of heads, viscera, frames, and skins, or others like tails, fins, scales, mince, and blood [21]. All this raw material can represent between 30 and 70% of the wet weight of the fish, depending on the species [35,48]. An estimate of the total generation of fish co-products using these estimates is presented in Figure 2. As an estimate, heads represent 9–12%, viscera 12–18%, skin 1–3%, bones 9%–15%, and scales 5% of total fish weight (Figure 3) [49,50], while the edible part of fish may represent 48–89%. Some of these co-products are considered easily degradable products (especially viscera and blood) given their high enzyme content, while others are considered to be more stable (e.g., bones, heads, and skin) [2].

### 2.2. Crustaceans

The global production of crustaceans is estimated to reach approximately 17 million tons worldwide, with aquaculture production (≈11 million tons) surpassing capture (approximately 6 million tons) (Figure 2) [29]. The whiteleg shrimp (*Penaeus vannamei*), the Chinese mitten crab (*Eriocheir sinensis*), and the giant tiger prawn (*Penaeus monodon*) are the most important crustacean marine/brackish–marine species produced in aquaculture in the world, while “Natantian decapods” as a group (including species of shrimp and prawns like, e.g., *P. vannamei* and *P. monodon*), Antarctic krill (*Euphausia superba*), and the Gazami crab (*Portunus trituberculatus*) are the most frequently captured species [29]. Crustacean processing, including various shrimp and crab species, produces multiple co-products, in this case mostly in the form of heads, pleopods, tails, and exoskeletons [52]. In general, it is estimated that up to approximately 75% of the total weight of crustaceans (e.g., shrimp, crabs, prawns, lobster, and krill) may become co-products in some form [53,54,55], which is an impressive value, potentially representing large environmental challenges locally. In the specific case of shrimp, the waste generated during industrial processing represents 40–50% of its total weight [56,57]. Shrimp co-products are generally discarded or processed into animal feed or protein feedstuff for aquaculture diets [58,59]. Specific shrimp co-products include the cephalothorax (the head and the thorax or pereon region) and the hepatopancreas (midgut diverticulum and primary digestive organ) as substantial non-edible parts [56,58], along with carapaces and tails [60]. Heads (cephalothorax) and tails are thought to be the richest in lipids [59]. In the lobster “tailing” process, the “head” (cephalothorax) is discarded as waste [61]. In the processing of shellfish, 50–70% is estimated to end up as co-products, including carapaces (namely heads), roe, and the hepatopancreas, which are also removed and traditionally discarded [62].

### 2.3. Mollusks

The production of mollusks for human consumption is estimated at a total of approximately 24 million tons worldwide (Figure 2) [29]. Here, aquaculture production (18 million tons) also surpasses the amount obtained by capture (approximately 6 million tons). Cupped oysters (*Magallana gigas* and *Magallana angulata*), the Japanese carpet shell (*Ruditapes philippinarum*), and scallops (including species from the *Placopecten* and *Pecten* genera) are the main mollusks produced in aquaculture and consumed worldwide [29]. Hard-shelled mollusk processing, including the handling of clams, mussels, and oysters for consumption, may generate attached soft tissues that can be exploited and repurposed. Undersized specimens are also usually discarded as waste [63]. Hard shells (of clams, mussels, and scallops) can account for 65–90% of live weight depending on the actual species [64,65,66]. The processing of cephalopods, such as octopuses and squids, is also a source of waste and co-products, corresponding to non-edible or less savory parts. Octopus processing originates co-products that may represent 10–15% of total weight after the commercial scission of tentacles and heads [67]. In the case of squids, the mantel is the portion directed for commercial purposes, with the heads, skin, viscera, tails, and ink becoming the processing co-products that are commonly discarded [68,69,70,71].

## 3. Marine Animal Co-Products as a Source of Healthy Lipids

The characterization of the fatty acid profile and the overall lipidome of marine animal co-products represents a starting point and a foundation for the ultimate valorization and rational utilization of these resources. Marine animal products, as foods, are commonly recognized for their beneficial features, especially their health-promoting profile in terms of lipid composition, displaying high polyunsaturated fatty acid (PUFA) content, along with low contents of saturated fatty acids and cholesterol [72]. One especially enticing characteristic is their high content of omega-3 PUFAs, particularly eicosapentaenoic acid (EPA) and docosahexaenoic acid (DHA) [73,74,75,76,77,78], which have been ascribed a vast list of health-promoting benefits [79,80,81]. But ultimately, it is the full spectrum of fatty acids, including monounsaturated and saturated fatty acids, which will determine the overall nutritional quality and even the stability of lipids [82]. There are many available reports characterizing the fatty acid profile of marine animal co-products, probably because of the aforementioned interest in omega-3 fatty acids, with the perspective of using these wastes as potential sources of highly valued fatty acids, such as EPA and DHA. There are fewer studies characterizing lipid class distribution in these matrixes, and just a few employing lipidomics and molecular-level profiling to characterize the lipid content in marine animal co-products. Anyway, it is well established that triglycerides and phospholipids are the main lipid classes present upon analysis of lipid extracts of marine animals [83,84,85,86,87,88,89], and that is also very generally the case for their co-products, as will be later detailed. Other lipid classes that were reported to be present in marine animal co-products and that will be appraised in this section include diacylglycerides and sterols, although the presence of lipid-soluble compounds like carotenoids and vitamins, which should also be present in lipid fractions, will also be highlighted and discussed.

### 3.1. Fish

Some lipid extracts of fish co-products have already been characterized for some marine species, but typically only on what concerns their fatty acid content. These characterizations reveal some variability in lipid content among different fish species and their co-products, but ultimately highlight the potential of these resources to represent a viable and significant source of healthy fats, with emphasis on omega-3 polyunsaturated fatty acids.

Fish nutritional composition varies with species, age, gender, health, nutrition and nutritional status, and seasonality [90]. Normally, and besides moisture (50–80%), fish are nutritionally rich in proteins (15–30% of total weight), although lipid content may be very variable (generally described as representing 0–25% of total weight) [91]. Depending on their fat content, fish may be classified as lean fish (cod (*G. morhua*) and hake (family Merlucciidae), presenting low lipid levels—0.5–3%), fatty fish, such as mackerel (family Scombridae) and salmon (*S. salar*) (10%–18% fat) [92], and semi-fatty fish, as gilthead sea bream (*Sparus aurata*), sea bass (*Dicentrarchus labrax*), and trout species, whose fat content in general is somewhere in between that of the former two [91].

A recent study encompassing several different co-products/parts (heads, frames, trimmings, skin, and viscera) of four of the most important marine fish farmed in Europe, Atlantic salmon (*S. salar*), European seabass (*D. labrax*), gilthead seabream (*S. aurata*), and turbot (*Scophthalmus maximus*), reported important lipid yields. For *S. salar* and *D. labrax*, viscera were the co-product displaying the greatest lipid yield (37.0% and 39.3% wet weight, respectively), while for *S. aurata*, the heads were the body part most abundant in lipids (22.3%); for *S. maximus*, the most lipid-rich part were frames (12.1%) [93]. Another study screening the lipid content of different fish co-product parts (head, gills, intestines, trimmings, bones, and skin) from meagre (*Argyrosomus regius*) and *S. aurata*, showed that the heads (28.9% and 37.1% DW, respectively), intestines (17.1% and 43.2% DW, respectively), and bones (35.0% and 30.6% DW, respectively), as well as the skin in the particular case of *S. aurata* (46.4% DW, only 6% DW in the meagre), were particularly rich in lipids [94]. Another study reported a 34.1% lipid content in the guts of *S. aurata* and 26.8% in its skin, which are levels too considerable not to be acknowledged [91]. The considerable lipid content in co-products of *D. labrax* were confirmed in another study [95], especially in the viscera (44.1% wet weight) and the liver (26.2%).

Lipid content in the co-products of aquaculture bluefin tuna (*T. thynnus*) was reported to represent up to 32.1% [96], while lipid content in the viscera of sardinelle (*Sardinella aurita*) was shown to vary seasonally, attaining a maximum yield of 25.4% (DW) [97]. However, other marine animal co-products were reported to display lower but still considerable lipid levels, such as the viscera of yellowfin tuna (*T. albacares*), containing 5.1% fat [98], the viscera of Atlantic herring (*Clupea harengus*), containing 9.6% lipids [99], or the viscera of salema (*Sarpa salpa*), displaying a maximum of 3.6% lipids [97]. Lipid content in another tuna fish (*Euthynnus affinis*) amounted to 7.0% in the head, 4.5% in the intestine, and 3.7% in the liver [100], while in sardines (*Sardinella lemuru*), fat content represents 5.7% in the head, 5.1% in the intestine, and 5.8% in the liver [101]. Lipids in cod offal represent 4.3% [102]. The lipid content in king salmon (*Oncorhynchus tshawytscha*) head, roe, and skin was reported to represent 35.5%, 13.4%, and 14.0% of wet tissue, respectively [103]. In blue mackerel (*Scomber australasicus*) processing co-products, namely in the head, skin, roe, and male gonads, the lipid contents were 12.3%, 20.9%, 9.4%, and 6.9% of wet tissue, respectively [104]. Regarding the lipid content of fishbones, the Pacific halibut (*Hippoglossus stenolepis*), the albacore (*Thunnus alalunga*), the sockeye salmon (*Oncorhynchus nerka*), the lingcod (*Ophiodon elongatus*), the golden pompano (*Trachinotus blochii*), the Atlantic cod (*G. morhua*), and the Chinook salmon (*O. tshawytscha*) have been reported to contain bone lipids in the range of 10–30% [105]. However, the total lipid content of fish bone powder from *Sardinella fimbriata* was relatively modest, estimated to be only 0.8% [106]. The Atlantic cod (*G. morhua*) is another species reported to have negligible bone lipid content (in the range of 1–3%) [105], with those values being comparable to those occurring in mammalian cortical bone tissue [107,108].

Most of the available studies characterizing the fatty acid profile of marine fish co-products focused on their fatty acid profile, certainly foreseeing the good nutritional characteristics normally assigned to marine fish products. Currently available studies are summarized in Table 1. Results are very variable, and this variability is certainly related to sampling specificities and to the specific technical approach applied in terms of extraction and fatty acid analysis. For instance, Soxhlet extraction seems to generate inferior yields in terms of EPA and DHA with regard to other approaches (Table 1). Nevertheless, it is fair to say that fish co-products generally present a significant content of PUFAs and omega-3, with the most consistent characteristic being the extremely low ratios of n-6/n-3 fatty acids, considered beneficial to health. It is also important to mention that, in some cases, co-products were analyzed in parallel with the actual edible fish parts (fillets) or whole fish, and in those cases, lipid profiles were very similar [91,109] or co-products even presented better yields [99], confirming the prospective quality of the fat present in this type of waste.

A few studies were able to characterize the lipid classes in terms of their distribution in several fish co-products. The phospholipid content in salmon heads was reported to represent 65.5% of total lipids, with the main polar lipids present being phosphatidylcholine (up to 43.4%) and phosphatidylethanolamine (31.2% of total phospholipid) [115]. Phospholipids in salmon (*S. salar*) heads were proposed to represent effective carriers of highly unsaturated fatty acids, given their remarkable content of EPA and especially DHA [113,115]. In the heads, roe, and skin of another salmon species (*O. tshawytscha*), phospholipids were reported to represent 1.9%, 1.3%, and 3.2% of wet tissue, respectively [103]. In all organs, phosphatidylcholine was the most abundant phospholipid class by a wide margin (especially in roe), followed by sphingomyelin in the head and roe, and phosphatidylethanolamine in the skin. Phospholipids were especially enriched in EPA and DHA when compared to neutral lipids, namely triglycerides [103]. The heads, skin, roe, and male gonads of blue mackerel (*S. australasicus*) were also studied in terms of their phospholipid content, displaying 2.7%, 2.8%, 3.5%, and 1.9% of phospholipids in wet tissue, respectively [93]. Again, phosphatidylcholine was the most abundant phospholipid class (especially in roe), with the exception of the gonads, where sphingomyelin was the major class present. Phospholipids containing omega-3 fatty acids were higher in roe (55.5%) compared to the head (40.9%), skin (21.8%), and male gonads (32%) [93].

### 3.2. Crustaceans

Crustacean processing co-products may represent viable and sustainable sources of quality lipids. The lipid content in the co-products of different species of crustaceans is very variable, depending on the species, the specificities of the co-products themselves, and the extraction methods employed for their characterization [116,117].

Co-products (heads, shells, and tails) of speckled shrimp (*Metapenaeus monoceros*) and striped prawn (*Penaeus kerathurus*) presented total lipid contents of 2.6% and 3.2%, respectively, which are higher values than those reported for the edible parts (muscle) [118]. Another study also reported that total lipid contents were higher in co-products of northern shrimp (*Pandalus borealis*) and the prawn *Trachypena curvirostris* (0.6% and 0.7% wet weight) compared to the muscle (0.4% and 0.3%, respectively) [57,59]. Brazilian redspotted shrimp (*Penaeus paulensis*) waste, also including a mixture of heads, shells, and tails, was reported to contain 4.9% lipids [116]. The lipid content in Argentine red shrimp (*Pleoticus muelleri*) co-products was reported to reach up to 11.3% (DW) in a mixture of carapaces plus heads (only 4% in just shells) [55].

Among shrimp waste products, shrimp cephalothorax and hepatopancreas have also been explored and proposed as viable sources of lipids with a high PUFA content. The total lipid content of brown shrimp (*Penaeus californiensis*) cephalotorax was reported to be 9.1% [119]. In Pacific white shrimp (*P. vannamei*), the cephalothorax and hepatopancreas presented lipid yields of up to 2.9% and 12.64% DW, respectively [58,120,121], although a different study reported a much higher lipid content in Pacific white shrimp cephalothorax (11.9% DW) [122]. In giant tiger prawn (*P. monodon*), the lipid content in the cephalothorax was reported to be higher in farmed (9.4% DW) than wild specimens (7.1%) [123]. Lipid content was higher in the heads than in the muscle of *P. monodon*, highlighting the quality of lipids from crustacean co-product sources. Also, total lipids in wild shrimp *P. kerathurus* cephalothorax (2.4% wet weight) was reported to be higher than those of muscle (1.0%) [124], and in Indian white shrimp (*Penaeus indicus*), total lipids in carapaces was shown to be 2% higher than those in the edible portion [125].

Among other crustaceans, the total lipid content in Norway lobster (*Nephrops norvegicus*) cephalothorax was reported to reach 11.5% (DW) in the summer [61]. Australian lobster (*Panulirus cygnus*) cephalothorax was reported to display a lipid content of 19.4% [126]. Lobster hepatopancreas was reported to be even more lipid-rich, with the lipid from these organs in Australian rock lobster (*Jasus edwardsii*) representing up to 24.3% [127]. For the side streams of the snow crab (*Chionoecetes opilio*), the reported lipid content was also very significant (14.8%) [128]. However, the southern king crab (*Lithodes santolla*) was reported to display a total lipid content of only 0.5% (DW) in its exoskeleton [55].

Most of the characterizations of lipid content in crustacean co-products focus on determining their fatty acid profiles (Table 2). Results vary, even for co-products from the same species, most likely due to the analytic techniques used. Nevertheless, in general, crustacean co-products do present significant amounts of PUFAs (close to or above 40%), low n-6/n-3 ratios, and, in some cases more than others, can represent viable sources of EPA and DHA. In some studies where co-products were studied in parallel to the edible parts of the animals (muscle), it is possible to observe that their fatty acid profiles are quite similar [57,118,124], highlighting the viability of using these co-products as sources of quality lipids.

A few studies characterizing the lipid fractions of crustacean co-products also discriminate the distributions of different lipid classes. Phospholipids are generally reported as being the major lipid class in crustacean co-products. Phospholipids (64.2% of total lipid) were in fact the main lipid class present in processing co-products from northern shrimp (*P. borealis*), with sterols representing 21.2% and triglycerides 13.7% [59]. In the cephalotorax of brown shrimp (*P. californiensis*), phospholipids were the main lipid class reported (6.0% of total weight), followed by sterols (2.7%) and free fatty acids (1.3%), with triglycerides only representing 0.6% of total weight [119]. The major lipid class reported to be present in the cephalothorax of Pacific white shrimp (*P. vannamei*) are also phospholipids, although their amount changes between reports (from 68.3% [121] to 82.51% [58]), with phosphatidylcholine and phosphatidylethanolamine being the main phospholipid classes present [134]. These phospholipid percentages are similar to those reported for the lipids in the meat of both the giant tiger prawn (*P. monodon*) and whiteleg shrimp (*P. vannamei*) (74.5% and 72.3%, respectively) [135]. The cephalothorax of *P. vannamei* contained 13.7% of phospholipids (DW), a significant part of which were PUFA-structured, and phosphatidylcholine was the most abundant class, followed by phosphatidylethanolamine [131]. The heads of *P. monodon* contained 39.2% phospholipids and 25.9% triglycerides (fewer phospholipids and more triglycerides than the muscle) [123]. Phosphatidylcholine represented 52.0% of total phospholipids, phosphatidylethanolamine 27.1%, sphingomyelin 9.0%, phosphatidylserine 7.9%, and phosphatidylinositol 2.9%, a profile very similar to the muscle [131]. In striped prawn (*P. kerathurus*) cephalothorax, polar lipids represented 48.3% of total lipids, with phospholipids representing 94.3% of the total polar lipid [124]. Similarly, in *P. monodon* cephalothorax [123], phosphatidylcholine was the major phospholipid present, representing 47.2% of total cephalothorax phospholipids, followed by phosphatidylethanolamine, representing 24.7% of polar lipids [124]. Phosphatidylethanolamine was found to contain higher proportions of omega-3 fatty acids than phosphatidylcholine, especially DHA [124], while the presence of plasmalogen species from both phosphatidylethanolamine and phosphatidylcholine was also unveiled [124]. Notwithstanding the studies referred above, the hepatopancreas of Japanese tiger prawn (*Penaeus japonicus*) displayed triglycerides as the main lipid class present (37.8%) and not phospholipids (35.5%) [130,136]. This was also the case in the hepatopancreas of *P. vannamei*, with triglycerides representing the major lipid class present, at 45.4%, and phospholipids representing 38.0% of total lipid content [58].

The lipid fraction of Norway lobster (*N. norvegicus*) cephalothorax was shown to contain 33.9% triglycerides and 31.8% phospholipids, with the other main lipid classes present being free fatty acids (15%) and cholesterol/sterols (14.9%) [61]. The main phospholipids present were phosphatidylcholine (14.0% of total lipids) and phosphatidylethanolamine (7.4%) [61]. Another study reported triglycerides to be the major lipid class in a homogenate of *C. opilio* co-products (more than 50% of total lipids), with phospholipids representing just 6.2% [128].

When comparing lipids from crustaceans to fish oils, one obvious difference is the presence of carotenoids as additional bioactive compounds [137]. Lipophilic fractions obtained from crustaceans (e.g., lobsters, shrimp, crabs, and krill) are an especially important source of natural carotenoids, with astaxanthin in particular being the most valued and most abundant [120,138,139,140]. Total carotenoid and astaxanthin content in crustacean co-products were shown to vary with species, season, and environmental grown conditions (feed and habitat) [62,141]. Shrimp waste (including from *P. paulensis*, *P. vannamei*, *P. borealis*, *P. indicus*, *Xiphopenaeus kroyeri*) has been reported to represent a good source of carotenoids, especially astaxanthin, presenting yields in the range of 40–148 µg/g waste [59,142,143,144,145,146]. Modern extraction approaches, namely ultrasonication/ultrasound-assisted techniques, have also been tested for the recovery of carotenoids from shrimp co-products with good results [132,133].

The exoskeletons of other crustaceans, namely crabs, were also reported to be good sources of carotenoids and astaxanthin in particular. The shells of the blue crab (*Callinectes bellicosus*) and the Mexican brown crab (*Callinectes sapidus*) contained 39 and 44 μg/g (DW) of astaxanthin [146], while the shell of the marine crab *Charybdis cruciata* contained 11.0 mg/g carotenoids, with astaxanthin representing 65.5% of that total [141]. Lobsters may also be an interesting source of natural carotenoids and astaxanthin, with yields of 70.4 and 41.6 μg/mL, respectively, being recorded in the exoskeletons of Australian rock lobsters (*J. edwardsii*) [62].

Liposoluble vitamins have also been described and quantified in crustacean co-products, such as the cephalothorax of whiteleg shrimp (*P. vannamei*), where vitamin A content ranged from 0.9 to 1.6 mg/100 g of waste, while vitamin E represented up to 49.0 mg/100 g [147].

### 3.3. Mollusks

Studies characterizing the lipid content of co-products of hard-shelled mollusks and cephalopods are scarcer, but the ones available also highlight the potential of these raw materials as sources of lipids with beneficial characteristics for human health.

Waste from great scallops (*Pecten maximus*) (comprising the mantle, gill, liver, digestive gland, and kidney) were reported to contain 8.6% total lipids (DW) in one study [148], while another reported higher levels (14.2% DW) [149]. The viscera of pen shell scallops (*Pinna rugosa*) presented 7.5% lipids in their composition,

With regard to cephalopods, the lipid yield of a pool of Patagonian squid (*Doriteuthis gahi*) co-products (viscera, heads, skin) was reported to amount to up to 10.3% (DW), depending on the extraction solvents used [68,150], while Argentine shortfin squid (*Illex argentinus*) viscera presented up to 11.1% lipid content [151] and the lipid content in giant squid (*Dosidicus gigas*) viscera was reported to represent 20.0% [119]. In viscera fractions from cuttlefish (*Sepia officinalis*), consisting of the stomach, intestines, and pyloric caeca, the lipid content was reported to represent 4.0% (DW) [97]. In the common octopus (*Octopus vulgaris*), co-products were reported to display a lipid content of 13.7% DW [67].

Focusing on the healthy characteristics of the fatty acid content in mollusk co-products, the available studies characterizing the fatty acid contents of these biological matrixes are summarized in Table 3. From the analysis of available information, it is possible to notice that mollusk co-products could generally represent excellent sources of PUFA and omega-3, with emphasis on both EPA and DHA. Moreover, in some cases, these enticing characteristics are comparable to those of the edible parts of these animals, namely in squid, thus confirming the viability of exploring these largely overlooked resources as healthy lipid sources [150].

Some studies characterizing lipid class distribution in mollusk co-products are also available. In co-products of great scallops (*P. maximus*) (a homogenate of the mantle, gill, liver, and kidney), phospholipids represented 1.8% of dry scallop waste [148]. Phospholipids were the most abundant lipid class in Patagonian squid (*D. gahi*) co-products (up to 46.4% of total lipids of pooled viscera, heads, and skin), followed by free fatty acids (up to 28.2%) and sterols (up to 13.2%), while triglycerides only represented up to 1.3.1% [150,152,153]. Another study reported the phospholipids in the co-products of *P. maximus* to be especially rich in DHA, with phosphatidylcholine being the major phospholipid class present, followed by lysophosphatidylcholine and phosphatidylethanolamine [149]. Considerable amounts of phosphatidylcholine and phosphatidylethanolamine plasmalogens were also detected. Also in the viscera of pen shell scallops (*P. rugosa*), phospholipids were again the main lipid class present (2.8% of total weight), followed by sterols (1.6%), with triglycerides representing 0.9% [119].

In turn, the viscera of giant squid (*D. gigas*) presented 10.8% phospholipids, 3.0% sterols, and 2.7% triglycerydes [119]. In Japanese flying squid (*Todarodes pacificus*) viscera residues, phospholipids were also the major lipid class present [154]. Phosphatidylcholine (80.5%) and phosphatidylethanolamine (13.2%) were the main phospholipids present, with both classes displaying significant amounts of EPA and DHA [154]. Pen squid (*Loligo* sp) and big blue octopus (*Octopus cyanea*) co-products were shown to be rather different, with sphingosines predominating in *Loligo* sp. extracts, while glycerolipids and glycerophospholipids predominated in *O. cyanea* [70]. Omega-3 fatty acids were reported to be major components of phospholipids, and substantial amounts of plasmalogens were also detected [70]. Phospholipids constituted 22.9% of total lipid content in the co-products of the common octopus (*O. vulgaris*) [67]. Interestingly, in other squid organs, namely the liver, the major lipid class were not phospholipids but rather triglycerides, as is the case for the arrow squid (*Heterololigo bleekeri*) (up to 63% of total lipids) [155], schoolmaster gonate squid (*Berryteithis magister*) (53%) [156], and Humboldt squid (*D. gigas*) (up to 27%) [157].

Patagonian squid (*D. gahi*) co-products were also explored for their liposoluble vitamin E contents in lipid extracts and were reported to be especially rich in tocopherols, (especially α-tocopherol, up to 2.8 mg/Kg), therefore being suggested as novel and valuable sources for α-tocopherol extraction from marine animal origin [68,150,153].

## 4. The Value of Marine Animal Co-Product Lipids for Human Health

Lipids make up a wide group of essential macronutrients which are incorporated into the diet and represent energy resources for the cell while also performing other roles in important physiological functions such as cellular signaling and regulation of cell membrane physical characteristics and function [158]. An unbalanced lipid intake can cause either a deficiency in essential fatty acids and fat-soluble vitamins [159,160,161], or, in excess, can lead to problems linked with metabolic syndrome, like hypertension, diabetes, and cardiovascular issues, but also to liver disease [162,163]. Recent knowledge has pointed out diacylglycerols and structured (and PUFA-rich) triglycerides and phospholipids (especially omega-3-containing phospholipids) as representing improved nutritional and health value [164]. The lipid content of marine animal co-products is generally recognized as being nutraceutically rich, including PUFAs, omega-3 fatty acids, and lipid-soluble vitamins. A summary of the beneficial health effects ascribed to the lipids present in marine animal co-products is presented in Figure 4. The lipid content in marine animal co-products, although not predominant, is still very substantial, especially, as described before, in certain fish and shellfish species (it can represent more than 25% of co-products of meagre (*A. regius*) [94], sea bass (*D. labrax*) [93,95], Chinook salmon (*O. tshawytscha*) [103], gilthead sea bream (*S. aurata*) [91,94], sardinelle (*S. aurita*) [97], salmon (*S. salar*) [93], or bluefin thuna (*T. thynnus*) [96]), crustaceans (it can amount to more than 10% DW in shrimp co-products of Argentine red shrimp (*P. muelleri*) [55] or whiteleg shrimp (*P. vannamei*) [132,133] and to more than 15% in some lobsters, namely Australian lobster (*P. cygnus*) [126] and Australian rock lobster (*J. edwardsii*) [127]) and mollusks (more than 10% in Patagonian squid (*D. gahi*) [68,150], Argentine shortfin squid (*I. argentinus*) [151], giant squid (*D. gigas*) [119], or common octopus (*O. vulgaris*) [67] co-products). Obviously, the higher the lipid content, the more justifiable it is to explore and take advantage of these co-product matrixes as possible sources of healthy lipids. Interestingly, the lipid content in some of the previously profiled co-products is comparable, or, at times, even higher than that reported in the edible portions of the same species; this feature has already been documented in fish [165], crustaceans [166,167], or mollusks [168]. A comprehensive study comprising fish, crustacean, and mollusk species showed that this is consistently the case, with co-products being more lipid-rich than edible muscle portions in all cases [169].

Lipids from marine animals are usually associated with a healthy content of PUFAs and omega-3 fatty acids, namely EPA and DHA [170], with DHA- and EPA-containing phospholipids being often referred to as “marine phospholipids” [171,172]. In fact, from a chemical standpoint, marine lipids are generally more varied in their fatty acid content and present longer chain fatty acids (besides the aforementioned prevalence of PUFAs, omega-3, and EPA and DHA in particular) than terrestrial plants and animals [173]. The fact that the human metabolism of alpha-linolenic acid of plant-derived origins to synthetize EPA de novo is negligible [79] and that the metabolism of EPA to DHA is virtually non-existent [174,175,176] implies that EPA and DHA must be acquired from the diet, making marine sources of these fatty acids even more critical from a nutritional standpoint. Moreover, the current human consumption and plasma levels of EPA and DHA are generally considered deficient, and are definitely lower than those considered to be ancestral values [177,178]. An intake of up to 0.5 g of EPA + DHA per day is recommended by health agencies to foster the prevention of cardiovascular diseases and other metabolic disorders [79,179,180,181].

The benefits of PUFA consumption have always been highlighted, normally in opposition to the detrimental effects of excessive saturated fatty acid consumption (namely increased risk of cardiovascular disease and type-2 diabetes) [182]. Lately, the benefits of PUFAs have been increasingly focused on the content of omega-3 fatty acids, and of EPA and DHA in particular [174,183,184,185]. Omega-3 fatty acids have essentially been elevated to the status of wide-spectrum nutraceuticals, with reported beneficial impacts on eye disease, bone health, fetal development, cardiovascular disease, diabetes, cancer treatment/prevention, cognitive function, neurodegenerative diseases, and inflammation [79,186,187,188,189,190]. The most immediate association between omega-3 fatty acids and health-promoting benefits concerns cardiovascular disease, although this association has been a matter of some controversy as of late [191,192,193]. Currently, there have been some steps towards the differentiation of the effects of EPA and DHA, with EPA being shown to present more beneficial effects than EPA/DHA formulations, at least for some specific cases [193,194]. In fact, the pharmacologies of EPA and DHA are distinct, with divergent effects on membrane structure, lipoprotein oxidation, and on the production of downstream metabolites that modulate the resolution of inflammation [194]. Marine animal co-products were reported to generally present substantial omega-3 fatty acid content, comparable to the edible parts of the animals, therefore making them good alternative sources of these specific nutrients.

Among fish co-products, those from marbled rockcod (*Notothenia rossii*, >30% omega-3 fatty acids [110]), blue mackerel (*S. australasicus*, >35% [104]), and Atlantic bluefin tuna (*T. thynnus*, 29.9% [96]), the viscera of sardinelle (*S. aurita*, 26.1% [97]), and the co-products of Atlantic herring (*C. harengus*, 26.5% [109]) are especially good potential omega-3 sources. In crustaceans, the exoskeletons of southern king crab (*L. santolla*, 40% [55]), exoskeletons and heads of Argentine red shrimp (*P. muelleri*, >40% [55]), processing co-products of northern shrimp (*P. borealis*, 37.1% [59]), and the cephalothorax of the Norway lobster (*N. norvegicus*, 27.6% omega-3 fatty acids) [61] present the highest percentages of omega-3. Mollusk co-products may represent especially promising sources of omega-3 fatty acids, with Patagonian squid (*D. gahi*) co-products containing up to 48.6% [150] and the co-products of great scallops (*P. maximus*) displaying a remarkable 40.7% of omega-3 fatty acids [149]. Concerning EPA, the processing co-products of *P. borealis* (21.1% EPA) [59] and the exoskeletons of southern king crab (*L. santolla*, 20.5%) [55] present the highest percentual EPA contents among crustaceans, as do the co-products of wild scallops (*P. maximus*, 20% [149]), Patagonian squid (*D. gahi*, 17.2% [150,153]), and pen shell scallops (*P. rugosa*, 17% [119]) among mollusks. As for DHA, the roe (and male gonads) of blue mackerel (*S. australasicus* [104]), shells and heads of Argentine red shrimp (*P. muelleri* [55]), and especially the co-products of mollusks, especially Patagonian squid (*D. gahi* [153]) and common octopus (*O. vulgaris* [67]), can contain over 20% of this particular omega-3 fatty acid. Another way to look at the benefits that lipids from marine animal co-products may entail has to do with the low n-6/n-3 fatty acid ratios consistently reported. A low n-6/n-3 ratio has been reported to promote beneficial effects on inflammatory conditions, cancer, and cardiovascular and neurological disorders [195,196]. A ratio of 4–5 to 1 or lower is recommended, although it is reported to normally be much higher in Western diets [197,198]. Therefore, with many co-products studied presenting extremely low n-6/n-3 ratios, they present nutritional characteristics that are optimal to mitigating the pernicious effects of modern diets in Western countries.

With the exception of a few specific cases mentioned before, like shrimp hepatopancreas (including *P. vannamei* and *P. japonicus* [58,130,136]), the cephalothorax of the Norway lobster (*N. norvegicus*) [61], or the livers of squids (*L. bleekeri*, *B. magister*, *D. gigas* [155,156,157]), phospholipids are generally the main lipid class in marine animal co-products. Like in the general case of phospholipids from marine sources, phospholipids from marine animal co-products are also rich in PUFAs, namely EPA and DHA [172,199], with these being mostly incorporated in the *sn*-2 chain [200]. Some reports state that omega-3 fatty acids bound to phospholipids are more efficiently absorbed and more efficiently delivered [201,202], outperforming triglycerides as omega-3 fatty acid carriers [171,203,204,205], which, in their turn, were suggested to be better carriers than omega-3 fatty acids in the ethyl ester form [80]. This would make formulations rich in “marine phospholipids” more valuable than common fish oils, where omega-3 fatty acids are present mostly in the triglyceride form, and ethyl esters to a lesser extent [80]. In agreement with this view, dietary approaches including omega-3 PUFA-structured phospholipids did, in fact, reveal an increased efficiency in the improvement of human health parameters when compared to commercial fish oils [171,199,206]. The fact that omega-3-containing phospholipids are normally ignored in the fish oil industry and frequently removed as an impurity during degumming processes [205] should be a matter of reflection and a driver for the reevaluation of how these resources are being exploited and valued. These characteristics of marine phospholipids are thought to be instrumental to the effects of omega-3 containing phospholipids in common features of aging and chronic diseases, as is the case of inflammation phenotypes, oxidative stress, neurodegenerative disease, and immune cell aging [115,202,207,208]. Moreover, phospholipids are pivotal for signal transduction in disease [209], and those including EPA and/or DHA have been specifically proposed to feature several health-promoting effects, namely counteracting cardiovascular disease, improving brain function and neurodegenerative conditions, presenting antitumor activity, and regulating lipid and glucose metabolisms [171,199,210,211,212,213]. Phospholipids have uses in the food manufacturing industry, namely as emulsifiers, antioxidants, and stabilizers [172,214].

Plasmalogens are a unique class of phospholipids, displaying a structure containing a fatty alcohol with a vinyl ether bond at the *sn*-1 position, while being enriched in polyunsaturated fatty acids at the *sn*-2 position of the glycerol backbone [215]. They are ubiquitous in animal membranes, both in invertebrates and vertebrates [216]. In the few studies that characterized marine animal co-product lipids at a molecular level, and particularly in mollusks [70,149], plasmalogens were described to be present in considerable amounts, specifically phosphatidylcholine and phosphatidylethanolamine plasmalogens. Plasmalogens were first proposed to be endogenous antioxidants and to be involved in membrane bilayer formation [217]. Lately, they have also been proposed to have a beneficial impact on atherosclerosis, on the prevention of inflammation (neuroinflammation in particular), on the improvement of cognitive function, and on the inhibition of neuronal cell death [217,218,219]. In fact, oral ingestion or plasmalogen replacement therapy were both put forward as novel strategies to target neurodegenerative diseases (namely Alzheimer’s disease) [217] and chronic inflammatory disorders [220]. Taking all this into account, marine animal co-products may indeed represent a convenient source of quality and healthy phospholipid and plasmalogen fractions that may find value in the food, supplement, and pharmaceutical industries.

Sterol content in marine animal co-products may be interesting from a valorization standpoint. In shrimp (*P. borealis* [59]), lobster (*N. norvegicus* [61]), and squid (*D. gahi* [150]) co-products, sterols were reported to represent a significant part of total lipids. This is particularly interesting since, at least in mollusks, anti-inflammatory activities of extracted sterols have been previously documented [221,222].

The content of carotenoids, and astaxanthin in particular, in the lipid fractions of crustacean co-products has been extensively explored and studied [146,223]. Astaxanthin, the main carotenoid present in these co-products, has been ascribed an especially potent antioxidant activity [224], but also other biological properties such as anti-inflammatory, antiproliferative, and anticancer activities [223,225]. Moreover, it has been proposed to present benefits in cardiovascular disease and inflammation contexts, while improving both lipid and glucose metabolism [223]. Therefore, this is another way to value the lipid fractions of marine animal co-products, especially crustaceans.

Marine animals, especially oily fish, have been reported to contain significant levels of tocopherol compounds [198,226]. In the case of marine animal waste, squid co-products have in fact been explored for their contents of liposoluble vitamin E, with interesting results [68,150,153]. Vitamin E is a chain-breaking antioxidant [68] and can exclusively be obtained from the diet [227]. It has been linked to many beneficial effects regarding, in general, conditions where oxidation plays a role, including cancer, aging, arthritis, and cataracts [228]. It has also been shown to be effective in the prevention of chronic inflammation and in the inhibition of platelet aggregation [229].

Finally, it is important to highlight that, more than just a potential to display beneficial effects based on their composition, some lipid fractions from marine animal co-products have in fact been ascribed interesting biological activities. Phospholipid extracts from shrimp heads, codfish roe, and squid gonads were tested for their antithrombotic, antistroke, anti-inflammatory, pro-angiogenic, and cardioprotective activities, with promising results [230]. Another study highlighted the anti-inflammatory activities of extracts from gloomy octopus (*Octopus tetricus*) viscera, squid (*Sepioteuthis australis*) heads, Australian sardine (*Sardinops sagax*) viscera/heads, salmon (*S. salar*) heads, and school prawn (*Penaeus plebejus*) viscera/heads [169]. Also, phospholipid extracts from the brain of skipjack tuna (*K. pelamis*) were shown to display macrophage-activating activity by inducing pro-inflammatory cytokines, therefore being suggested as possible boosters for human immunity [231]. Lipid extracts from different organs (stomach, liver, brain, and skin) of marbled rockcod (*N. rossii*) and mackerel icefish (*C. gunnari*) were tested for their potential in skin protection, with promising results [110]. Also, a phospholipid-rich extract from salmon (*S. salar*) heads was shown to elicit favorable effects in rat models of metabolic syndrome [115]. Moreover, an acetone extract of northern shrimp (*P. borealis*) industry processing waste was reported to display neuroprotective effects via antioxidant and anti-inflammatory effects and by increasing neurotrophins [232]. A lipid extract from Pacific white shrimp (*P. vannamei*) cephalothorax was shown to display antioxidant and anti-inflammatory activities, especially when encapsulated by spray-drying [233], while its hexane extract was shown to present significant antibacterial activity [234]. A lipid extract from a *P. borealis* processing co-product obtained using Soxhlet extraction showed the potential to elicit antiadipogenic effects [59]. Phospholipids from yet another shrimp co-product (heads of *P. vannamei*) showed angiogenic, antithrombotic, antiarrhythmia, and anti-inflammatory activities in zebrafish models [134]. Regarding mollusks, lipid fractions of common octopus (*O. vulgaris*) co-products (viscera, ink sac, eyes) obtained using traditional methods (Folch extraction) were shown to possess antiproliferative and apoptotic effects on human breast cancer cell lines [235]. Finally, cupped oyster (*M. gigas*) co-product extracts were shown to decrease lipid cholesterol and triglyceride content in rat livers and were suggested as potential lipid-lowering functional foods or supplements [236,237]. A summary of the reported biological activities of lipid fractions of marine animal co-products is depicted in Figure 4.

## 5. The Value of Marine Animal Co-Product Lipids for Various Industries

Given their enticing chemical qualities in terms of composition, marine co-product lipids may find their way into diverse applications in the industry. The first and most obvious application would be the incorporation of marine animal co-product lipids into food products, enhancing their nutritional profile while also creating innovative and health-focused products that cater to evolving consumer preferences and dietary trends. In “functional” or “designer” foods, conceived to enhance human health and wellbeing, marine co-product lipids offer the possibility of enriching diets in omega-3 fatty acids and other bioactive compounds, while also improving their sensory attributes. The most obvious commercially available products that are fortified in omega-3 fatty acids are dairy products, namely yoghurt, milk drinks, margarines, spreads, and fresh and ultra-high-temperature milk [238,239]. However, there are also less evident examples of omega-3-enriched foods, such as meat, eggs, baked goods, beverages, and even infant formulas; these products are generally marketed for their cardiovascular benefits and brain-boosting properties [240,241]. The use of omega-3 fatty acids derived from marine animal co-products should be enticing given the significant proportion of these fatty acids in these products in general, and should currently only be limited by further prospection and characterization efforts and regulatory restrictions. However, there are already examples of the incorporation of oils from marine animal co-products into foods, such as baked goods, dairy, and meat products [242]. In the food industry, marine animal fats have also been explored as additives to improve the characteristics of salad dressings and mayonnaise [243] and yoghurt [244], namely their oxidative stability. A specific case of a concrete application of marine animal co-product lipid extracts in the food industry is the suggested incorporation of shrimp cephalothorax lipid extracts into food products, including soups, sauces, and meat or fishery products, where it would function as a food coloring agent and a functional ingredient [233,245,246]. In fact, these shrimp lipid extracts display interesting anti-inflammatory and antioxidant activities, significant coloring capacity, and relative stability under thermal treatment and refrigerated storage, therefore presenting intrinsically valuable qualities for the food industry [233].

Taking into account their qualities, especially their abundance in omega-3 fatty acids, marine animal co-products also appeal to companies working in the field of nutraceuticals, and even more so in the recent field of functional lipids. Omega-3 supplements are commercially available in a variety of different formulations from different brands, mostly as (fish) oils or as capsules, which are thought to promote an increased shelf life and improve absorption times [247,248,249,250]. As an alternative to the direct commercialization of marine oils, formulations (concentrated or nanoliposome-containing) of EPA and DHA specifically can also be used as supplements in human nutrition [251,252]. The global lipid nutrition market, mostly based on the promise of omega-3 fatty acids as health promoters, is expected to reach USD 17 million by 2031, growing by 7.6% annually over the 2021–2031 period, driven by increasing demand [253]. Therefore, the market for additional healthy lipid sources actually exists, and the use of omega-3 fatty acids derived from marine animal co-products should only depend on further technical advances for the optimization of extraction yields using green approaches and regulatory adjustments.

Phospholipids are part of the EU’s list of authorized food additives (E322 lecithin), and are used as emulsifiers and antioxidants in foods [254]. Phospholipids have in fact been ascribed antioxidant properties during food processing, depending on the amine composition of their head group and on their fatty acid composition [255,256]. Phospholipid sourcing is, nonetheless, limited. Food lecithins (basically a complex mixture of phospholipids from natural sources) are typically produced from oil-degumming pastes (soy, sunflower, and, more recently, rapeseed) and egg yolk [186]. The fatty acyl compositions of the side chains of the phospholipids present in lecithins are relatively simple, not very varied, and display a low degree of unsaturation [257]. More recently, krill oil has been explored as a marine animal source of omega-3 rich phospholipids [201]. However, given the promising perspectives of their use and the pressure created by increased demand [258], marine animal co-products, generally presenting phospholipids as the most abundant lipid class, may provide a viable, still largely unexplored source of these compounds.

Lipids from co-products of marine animals may also be appealing to the fields of pharmacology and drug development. As previously mentioned, phospholipid extracts from co-products derived from fish, crustaceans, and mollusks have been shown to display interesting biological activities, and have therefore been suggested as possible targets for pharmacological and clinical studies and development, namely in the context of inflammatory and cardiovascular diseases [230]. Moreover, phospholipids have also attracted interest as drug delivery systems on the basis of their excellent biocompatibility and amphiphilicity [259,260]. Liposomes, intravenous lipid emulsions, micelles, drug–phospholipids complexes, and cochleates are all phospholipid-based delivery vectors [260]. Liposomes, in particular, have particularly been used as delivery systems in food, cosmetic, and pharmaceutical applications [261]. Given their specific characteristics, namely their high contents of omega-3 fatty acids, it has been suggested that marine phospholipid liposomes could promote an enhanced bioavailability and activity of encapsulated functional compounds [171]. The exploration of marine animal co-products as viable and sustainable sources of marine phospholipids with benefits for the pharmaceutical industry is currently unexplored. In fact, only a limited number of reports exist characterizing and quantifying marine phospholipids in marine animal co-products. Further lipidomic characterization of these matrixes, along with the development of green strategies to isolate these compounds, will help to signal preferentially rich sources and determine the viability of exploring these resources for marine phospholipid isolation.

In the specific case of plasmalogens, they have also garnered interest from a pharmacological perspective, having been proposed to display attractive functionalities as healthcare materials. In fact, they have been proposed to present promising characteristics to serve as constituents of functional membranes of biosensors, light-activated liposomes, or nanoparticles with endosomal escape capabilities [219]. Moreover, marine organisms have also been proposed as alternative sources of complex lipids (as plasmalogens) as an alternative to more common sources, such as bovine brains, whose use is now unwarranted due to past outbreaks of bovine spongiform encephalopathy [219]. Currently, a possible significant use of marine animal co-products as sources of plasmalogens with interest for the pharmaceutical industry is mostly hindered by the lack of available information and proper in-depth lipidomic characterization of the content of these resources at a molecular level.

Finally, lipids from marine animal co-products may also be appealing for the cosmetics and personal care industry. In fact, marine animal-derived lipids were shown to display good conditioning, moisturizing, and emollient abilities [262,263]. Moreover, fish oils, some of which are produced using seafood industry co-products, have been ascribed broad benefits for the maintenance of skin homeostasis as well as in skin disorder contexts, including photoaging, cutaneous carcinogenesis, dermatitis, cutaneous wounds, and hyperpigmentation [264]. The skin health-promoting characteristics of these oils are generally related to their content of omega-3 fatty acids, especially EPA and DHA [264]. This evidence should justify further interest in studying the potential and activities of lipid fractions from marine animal co-products in skin care, and therefore uncover additional value for the cosmeceutical industry. Astaxanthin, in particular, has also showed promise for cosmetics, mainly because of its remarkable antioxidant and radical scavenging abilities [145]. These properties also justify the interest in exploring the use of astaxanthin in nutraceutical and healthcare applications [265,266]. Finally, tocopherols obtained from shrimp and squid co-products should also gather obvious interest from both pharmaceutical [267] and cosmeceutical [268] industries.

There are several compounds that are extracted from marine animal co-products with established demand for high-end uses in the pharmaceutical and cosmeceutical industries. Collagen, gelatin and collagen derivatives [269,270], protein hydrolysates [269], chitin and chitosan [269,270,271,272,273,274], glycosaminoglycans [250,275], and hydroxyapatite [276,277], in particular, have a well-established importance in the pharmaceutical, cosmeceutical, and biomedicine industries (in addition to the aforementioned astaxanthin). The prospection of bioactive lipids and the investigation of the biological activities of lipid extracts from marine animal co-products have the potential to add to this list and to increase the value of these resources, expanding their applications for higher-end purposes.

Aquaculture systems are absolutely reliant on the production of feeds of both marine and terrestrial origin, with feed production being pointed to as the most significant source of environmental impact on fed aquaculture production [278]. Other than their documented benefits when consumed by humans, DHA and EPA omega-3 PUFAs are essential ingredients with high demand in aquaculture (namely of marine species), where they promote the growth and overall health of farmed animals [279,280]. This is another justification for the increased demand for omega-3 PUFAs, putting pressure on supply through conventional sources [281,282]. Therefore, new omega-3 sources are needed to alleviate the pressure of increasing demand for low-environmental-impact feeds assuring the quality of aquaculture animals. The repurposing of marine animal co-products aligns with eco-intensification paradigms and can promote a lower Fish In: Fish Out (FIFO) ratio [283]. In fact, oils produced from marine animal co-products (e.g., tuna fish side streams) have been ascribed beneficial effects when used in formulations, preventing excessive fat deposition in farmed fish [96]. Also, products based on processed side streams of shrimp and crab are already being used in animal feeds to balance their nutrient profiles [284]. Other lipophilic compounds present in the lipid fractions of marine animal (in this case, crustacean) co-products have also garnered particular interest in the aquaculture field, with astaxanthin being approved by US and EU authorities to be used as a colorant/dyeing agent in animal feed and fish food, salmon in particular [145,285,286].

## 6. Sustainability and Environmental Impact

Every year, the amount of waste from the world’s fisheries exceeds 20 million tons [287]; therefore, the disposal and recycling of such large amounts of biomass represents a challenging task. Measures like the recently enforced Landing Obligation of the European Common Fisheries Policy show a trend for legislators to aim at mitigating environmental impact of fishing operations, but also imply more costs for the disposal of those additional specimens, which are now not permitted to be returned to sea [288]. Therefore, repurposing marine animal food co-products should represent a justified sustainable and environmentally responsible practice, which aligns with the principles of a circular economy [34] and with the United Nations’ Sustainable Development Goals [289].

The exploitation of the use of marine animal co-products in a “Waste to Wealth” approach, representing economic, environmental, and food security benefits, has been widely explored [24]. However, the strategic management of marine animal co-products must further employ the concepts of a circular economy and life cycle thinking in order to increase their efficient use and mitigate the environmental impact of the seafood industry [34]. Marine animal co-products are generally recognized for having an interesting composition from a chemical standpoint, including valuable protein and lipid fractions, minerals, enzymes, and vitamins [250]. Currently, there are several main paths established for the generation of value from marine animal co-products: the acquisition of marine proteins (fishmeal, silage and hydrolysates), the production of PUFA-enriched oils, the production of biodiesel and biogas, and the isolation of higher-end compounds, such as vitamins, enzymes, minerals, taurine, creatine, and hydroxyapatite, directed for specific industrial or pharmaceutical uses [290]. A more efficient and profitable use of marine animal co-products depends on new valorization approaches and improved recovery technologies for the already signaled and novel compounds of interest detected [33,291], and in this case, the specific presence of bioactive lipids has been patently disregarded.

The identification and exploration of new healthy lipid sources is particularly important as the global population keeps growing, and sustainable food supplies become more critical [292]. A more rational use of marine animal co-products will reduce the waste generated by seafood processing by assuring that a larger portion of harvested marine animal biomass is utilized. This optimization of the use of marine animals from capture or farming can help to alleviate the pressure upon wild animal populations and contribute to the overall conservation of marine ecosystems [34,293]. Moreover, it can contribute to more eco-friendly seafood processing practices, allowing us to also aim at broader environmental goals, such as reducing the industry’s carbon footprint and greenhouse gas emissions and alleviating the burden on landfill sites. In fact, in most instances, marine animal co-products are incinerated, composted, anaerobically digested, landfilled, returned to the sea, or even simply abandoned [22,294], with consequent negative ecological and human health impacts. However, from both ecological and economic standpoints, the valorization and utilization of marine animal co-products for the collection of valuable compounds should always represent a preferential approach compared to all other options [295]. Interestingly, downstream from the farmed animals themselves, effluents from the processing industry (fish canning in particular) were also explored as sources of omega-3 fatty acids [296]. This is a new perspective on the valorization of lipids from seafood processing that could also contribute to sustainability and reduce the environmental impact of such enterprises, increasing circularity and framing these economic activities under a bioeconomy paradigm.

The repurposing of marine animal co-products may also represent an extra revenue source for seafood processors and related industries, an opportunity to foster innovation in the form of novel value-added products, and diversification within the seafood industry with additional eco-friendly options. It also aligns with the broader goals of sustainable seafood certification programs and fishery management initiatives [297,298]. Moreover, the exploration of these resources may represent a boost for local economies and lead to the creation of new jobs in the seafood industry. Ultimately, the incorporation of marine co-product lipids into the seafood value chain contributes to a more responsible and environmentally conscious, effective, and productive seafood industry.

Obviously, the extraction and valorization of lipids from marine animal co-product matrixes may not be appropriate or possible in all cases, namely in the case of matrixes with poor lipid yields. However, for co-products with more promising features, lipid extraction should be explored in adequate and suitable frameworks. Lipid extraction could be included in a rational biorefinery pipeline approach, incorporating energy-efficient techniques, waste reduction, and recycling, making it attractive and economically sustained. There are already many individual instances of established high-value compounds being effectively harvested from marine animal co-products, such as collagen and gelatin, biopolymers like chitin or chitosan, hydroxyapatite, carotenoids, pigments, proteins and protein hydrolysates, and bioactive peptides and minerals [21,34,35,36,37,52,299,300,301]. Therefore, it may be feasible and even advisable to envision efficient biorefinery strategies to maximize resource utilization, ensuring that each type of co-product is utilized to its fullest potential and that all valuable compounds are accounted for and collected, while also minimizing costs and waste. There are several different approaches related to how lipid extraction could be incorporated in such procedures, and they would always depend on the specific biological matrix/co-product being targeted, on the presence of compounds of interest, and on the specific needs of the market. In any case, lipid extraction could represent a key extraction step or a common step alongside other existing/well-established processes, such as collagen, chitin and chitosan, and protein and protein derivative extraction, depending on the source co-product and on the intended outputs of the biorefinery pipeline (Figure 5). At the end of the process, recovered lipids could then be directed into various value chains depending on their quality and composition, with the use of high-quality lipids being advocated for the production of functional foods, nutraceuticals, and dietary supplements, while lower-grade lipid extracts, or those co-products failing the requirements to maintain “food quality”, could be aimed at incorporation into feeds or even biofuel or biogas production [93,299,302,303]. Such pipeline approaches have already been proposed in the case of shrimp exoskeletons, although they did not encompass lipid recovery [304,305], and in the case of fish waste, with the production of fish oils as an integrated step [306]. This inclusion in biorefinery platforms, aiming at extracting all possible compounds representing value, aligns very well with the “Zero Discards” mandate included in the Sustainable Developmental Goals of the United Nations, as well with the EU policy agenda, which includes the Circular Economy Action Plan [307] aiming to reduce raw materials and associated environmental pressures, the Bioeconomy Strategy [308] targeting the exploitation of biomaterials in a sustainable manner, as well as the European Biorefinery outlook to 2030 [309].

## 7. Challenges and Future Directions

Some of the challenges underlying a sustainable and profitable exploration of lipids derived from marine animal co-products have to do with the lack of available information. When writing the present review, other than estimates, it was challenging to find credible information on the true numbers of marine animal co-products produced at the global level, nor was it easy to find systematic studies calculating the carbon footprint and actual costs of the disposal of such biomass by the industry [310]. Both types of information would likely reinforce the need to take further action to minimize the waste that is still associated with the use of marine animals and their co-products, as well as the potential economic upside that the full use of these resources may represent, if rationally perceived and explored. Still on the topic of the lack of available information, and despite the many works available on the characterization of lipid fractions in marine animal co-products, especially in terms of their fatty acid profiles, there is still a lack of information on their lipid composition at the molecular level. Few comprehensive studies have addressed the structural characterization of these biological matrixes, and such studies are paramount to better understand how important fatty acids of marine origin to which important beneficial health effects (omega-3, EPA and DHA in particular) are ascribed are distributed along lipid classes, since this has major implications for their absorption and general bioavailability. Moreover, there are only a few studies scanning the biological activities of lipid fractions from marine animal co-products, although the ones available have shown promising results. Identifying extracts displaying biological activity, and further investigating their composition and identifying the active lipids involved in the biological effects, would be another practical possibility to foster the valorization of these co-products. This would pave the way to possible high-end applications in the pharmacological and cosmeceutical industries. The main bottleneck is that, as of now, the mixing of co-products is still a common practice, as could be perceived in several characterization studies surveyed in this review; this practice hinders the discovery of specific lipid agents that are potentially nutritionally superior or display particular bioactivities, and consequently limits their downstream applications.

There are also some technical problems related to the processing of marine animal co-products that may hinder the collection of the lipids of interest. Marine animal co-products are generally highly perishable materials. These co-products are water- and nutrient-rich, thus meeting the basic requirements for good growth mediums for fungi and bacteria [22,311]. This means that, if not processed quickly, storage may result in the loss of raw material and nutritional value and in the oxidation of the compounds of interest (such as carotenoids and PUFAs, which are highly prone to lipid peroxidation) [58,312,313]. In fact, PUFAs are readily degraded by lipid oxidation reactions into a myriad of secondary oxidation products, of which short-chain saturated and unsaturated carbonyl compounds (including both aldehydes and ketones) are supposedly the ones contributing the most to flavor deterioration and the occurrence of off-flavors evoking fishy, metallic, and rancid sensations [314,315,316]. Curiously, it was described that off-odors elicited by the oxidation of PUFAs depend on the specific composition of fatty acids, with different proportions of EPA and DHA modulating the sensory profile [317]. Taking into account the amounts of waste estimated to originate in the seafood industry, the constraints imposed by the perishable nature of these resources could pose an obvious problem of scale that would have to be considered from the operational and financial standpoints. The option to, in some cases, store these co-products at low temperatures at integrated centralized facilities, could be advantageous, as small/artisanal fisheries, potentially unlike bigger operators, do not have the means to store and process these co-products. Centralized facilities may also be justifiable in the context of implementing biorefinery pipeline approaches.

One significant challenge for the widespread and profitable use of marine animal co-products lies in the technical aspects related to the lipid extraction itself. In fact, as of now, the standardization and optimization of green extraction procedures, guaranteeing quality and yield, are still a work in progress [318,319,320]. Despite the availability of many different extraction techniques, selecting the most suitable method for a specific co-product remains a complex task, taking into account the heterogeneity and specificities that these co-products may present. Moreover, factors like cost-effectiveness, energy efficiency, and scalability must be carefully considered when appraising each possible approach. Solvent-based extraction is the most commonly used methodology for lipid extraction, especially procedures based on the Bligh and Dyer [321] or the Folch [322] methods. Although these are very efficient methods, both in terms of lipid yield and lipid classes covered, making them very convenient for characterization purposes, the use of noxious chemical solvents renders them unsuitable for the use in food/feed industries or for any other application related to direct human consumption; thus, there is a requirement for the use of food-grade and non-toxic solvents. Therefore, modern green lipid extraction techniques, compatible with human use, should preferentially be explored and applied in the processing of marine animal co-products with the objective of maximizing applicability. According to Directive 2009/32/EC of the European Parliament and Council of 23 April 2009 on the extraction solvents used in the production of foodstuffs and food ingredients, organic solvents currently allowed in the industry include propane, butane, ethyl acetate, ethanol, and acetone [323]. Some ionic liquids, including deep eutectic solvents, are also considered green solvents [324]. Therefore, all these could be potentially suitable alternatives to be included in lipid extraction strategies from marine animal co-products. Among some of the innovative green technical approaches that may be employed to obtain the lipid fractions of marine animal co-products, we may mention supercritical fluid-, enzyme-, microwave-, and ultrasound-assisted extraction techniques [22,325,326,327,328]. In recent years, supercritical fluid extraction has gained traction as a leading option for lipid extraction from different biological matrixes, including marine animal co-products, presenting a good efficacy in the recovery of omega-3-rich marine oils [290,329,330,331]. This technique presents operational conditions that are more favorable from an environmental and industrial processing viewpoint, such as sparing the need to use high-temperature treatments that can lead to lipid oxidation/degradation or the use of organic solvents [247,290]. However, supercritical extraction is more directed towards the isolation of neutral lipids, meaning that it preferentially targets the extraction of triglycerides; nevertheless, the use of modifiers, such as ethanol, may also render this technique applicable to the extraction of phospholipid rich-extracts [63,149,332]. However, in the end, the cost of adapting the application of these techniques to an industrial scale seems to represent the ultimate challenge for their use [290]. As such, understanding the nuances of all these extraction techniques, enabling the use of sustainable and green solvents, and optimizing and tailoring them to specific marine animal co-products are essential for devising integrated approaches optimizing yield, quality, and profit. The establishment of a strong engagement between research and industry will be pivotal for the development and success of such strategies.

Another technical issue that cannot be overlooked has to do with interindividual variability in animals, which may be raised by factors such as age, sex, environmental conditions, season, nutrition, and processing [333], and ultimately may generate fluctuations in the yield and composition of specific co-products. Other technical issues are related to the stability of the biomolecules of interest along the extraction/biorefinery processes, which should be studied and assured. Other than that, it may also be necessary to develop thorough studies regarding sensorial and bioavailability aspects, as well as possible interaction with other ingredients when incorporated in formulations.

Moreover, another technical challenge may have to do with the potential presence of contaminants in marine animal co-products, namely those known to accumulate in marine animal biomass, such as metals (and metalloids), pesticides, polychlorinated biphenyls, dioxins, and (micro)plastics [126,283,325,326,327]. Contaminant accumulation has been reported in crustacean hepatopancreas [334,335,336] and in the skin, viscera, bones, and scales of fish [337,338,339,340,341]. Mollusks also accumulate toxicants [342,343,344,345], with the digestive glands and gills of hard-shelled animals representing preferential accumulation sites [346,347,348]. The presence of marine biotoxins may represent a very concrete challenge for the use of marine animal co-products. These biotoxins, produced by bacteria, cyanobacteria, and microalgae, are known to bioaccumulate in fish, mollusks, and crustaceans [349,350,351,352,353]. In the case of the presence of biotoxins in fish co-products in particular, this may in fact represent a problem, since fish appear to preferentially accumulate toxins in the viscera rather than in the flesh [354,355,356,357,358]. Therefore, it becomes paramount to signal the collection of fish and marine animals in the vicinities of harmful algal blooms where these toxins are abundantly produced [359], even for purposes of making use of animal parts not directly intended for dietary consumption. These concerns about the presence of contaminants in marine animal co-products should not be taken lightly and may imply additional thorough quality control measures and investments in hazard analysis and critical control point (HACCP) and decontamination procedures, ensuring the safety and purity of lipid extracts.

Regulations regarding the disposal of marine animal co-products by fisheries or the seafood industry can be very uneven worldwide, and conflicting regulatory and economic drivers often create perverse incentives leading to practices that are not ecologically desirable [360]. In the European Union, restrictions on the use of animal co-products were codified in 2001 through the adoption of Regulation (EC) No 999/2001 [361]. Moreover, EU Regulation 1069/2009, as implemented by EU Regulation 142/2011 [362], governs the collection, transportation, storage, handling, processing, and use or disposal of all animal co-products, including fish material not destined for human consumption, finfish processing co-products, and shellfish surpassing shelf life [363]. This legislation includes safety regulations and sanitary practices that must be followed to ensure the suitability of co-products for various uses. The European Union is also actively enforcing policies to actively promote food and seafood circularity, namely within the scope of the Circular Economy Action Plan framed within the European Green Deal [364]. However, available legislation must of course be dynamically adjusted in order to accommodate innovations regarding co-product processing and eventual novel products and processes that may arise from these applications. Other changes could contemplate introducing more accuracy and transparency in the reporting/communication of the amount of waste generated by the industry and the associated costs of disposal, which would be very informative in the contexts of appraising environmental impacts and promoting a more efficient use of these co-products as largely untapped resources. More than just governing the handling and disposal of marine animal co-products, regulatory considerations will also inevitably shape their utilization. Ensuring compliance with food safety and labeling regulations is critical for generating trust when these materials are eventually incorporated into functional foods and nutraceuticals. This is a task that requires a strong synergy and cooperation between research, industry, and legislators. For this purpose, a reliable traceability system is paramount. Moreover, efficient marketing strategies and consumer acceptance may have to rely not only on nutritional attributes but also on the certification of ecologically conscious, sustainable practices to engage stakeholders, particularly consumers.

## 8. Conclusions

This review attempted to present an integrated perspective on the potential use of marine animal co-products as sustainable sources of health-promoting and bioactive lipids. These lipid sources remain significantly undervalued and their full valorization may generate both financial and ecological benefits. A summary of the advantages of further exploring these resources in terms of their lipid content is depicted in Figure 6.

Marine animal lipid co-products have been adequately characterized in terms of total lipid content and fatty acid profiles, but less thoroughly in terms of lipid class distributions and even less so at the molecular level (e.g., by using mass spectrometry-based lipidomics approaches). However, despite the need for more in-depth characterization work, most of the characterized co-products show very favorable features in nutritional terms, although they may still require processing steps to transform them into attractive nutritional products. Ultimately, these characteristics should be very appealing for the food and supplement industries, as well as for feed applications. However, looking more mindfully at lipid compositions and the available bioactivity studies, it is fair to foresee novel ways to add value to these marine resources and repurpose them for more high-end applications, namely in the pharmacological and cosmeceutical industries. Moreover, the sustainability aspect of valuing marine animal co-products cannot be overstated. On the one hand, by optimizing the use of edible marine animals as fundamental resources, we are contributing to the conservation of marine ecosystems and to a more sustainable approach towards the capture of these animals from the wild, as well as their aquaculture. By diverting these often-discarded materials towards added-value applications, we are minimizing waste generation and the ecological toll that the seafood industry still has on the world’s oceans and seas. Obtaining health-promoting lipids from marine animal co-products could be integrated with the extraction of other more established compounds of interest in a centralized biorefinery pipeline approach, under strict eco-efficiency and eco-design principles, limiting the costs and optimizing the output of these resources. As the demand for sustainable and health-focused ingredients continues to rise, the full use of marine co-product lipids presents a promising and sustainable solution, bridging health and environmental goals. Therefore, and despite some prevailing challenges regarding technical optimization and regulatory compliance, the future use of lipids derived from marine animal co-products is certainly an endeavor worth pursuing, ultimately contributing to addressing societal challenges and to a more sustainable blue economy.

## Figures and Tables

**Figure 1 marinedrugs-22-00073-f001:**
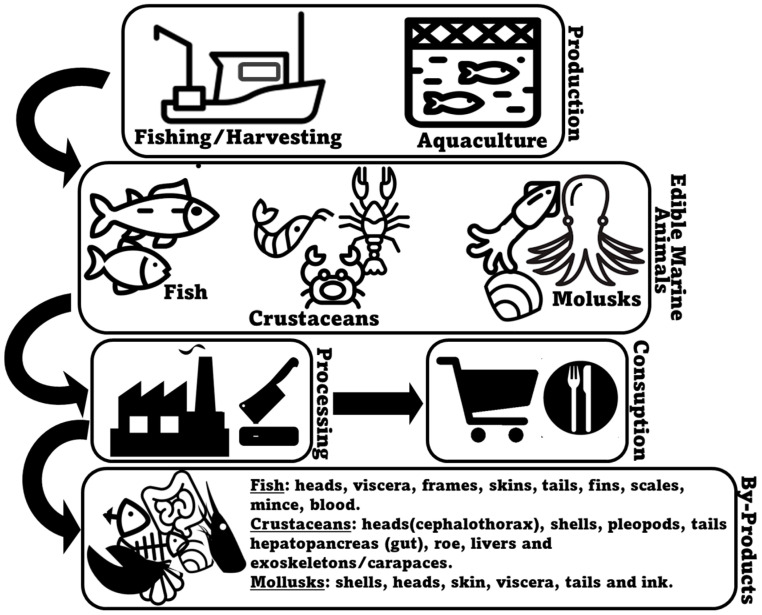
An overview of the generation of marine animal co-products.

**Figure 2 marinedrugs-22-00073-f002:**
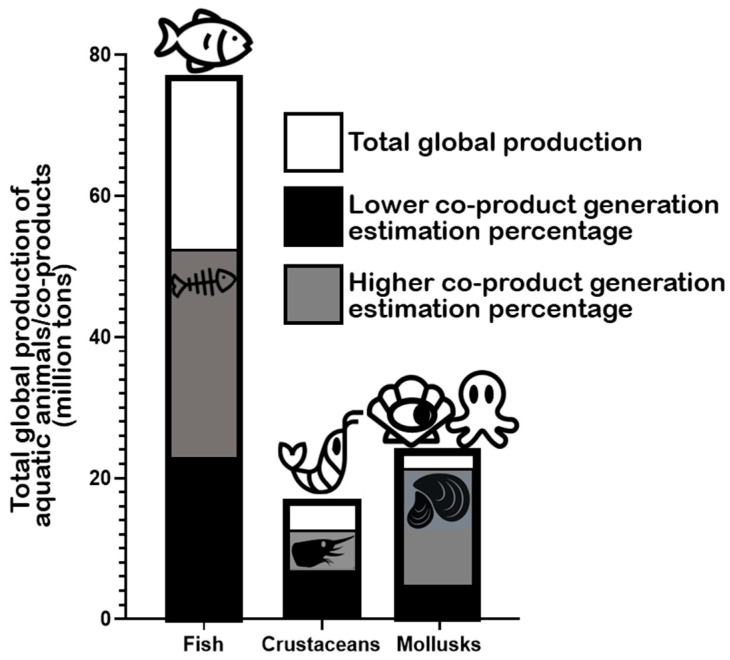
Total global production of fish, crustaceans, and mollusks according to the “The State of World Fisheries and Aquaculture” (2022) by FAO and estimated generated co-products according to generally accepted estimates (mentioned in the text).

**Figure 3 marinedrugs-22-00073-f003:**
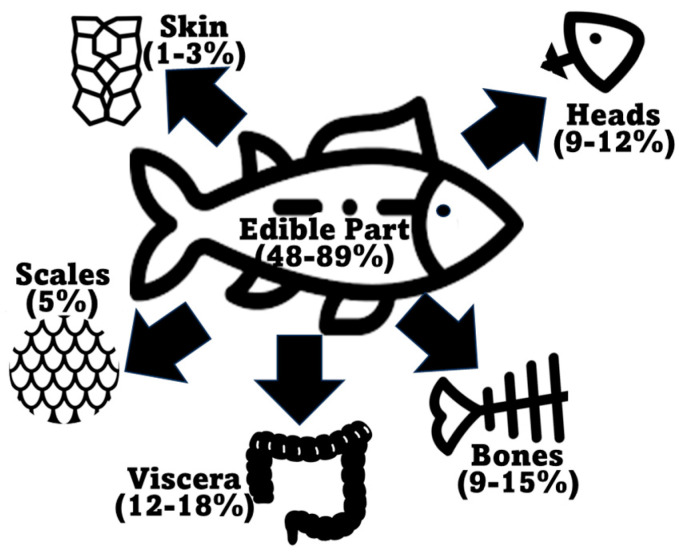
Estimation of generated fish co-product percentages (according to Pedrosa et al., 2014 [51], and Boronat et al., 2023 [50]).

**Figure 4 marinedrugs-22-00073-f004:**
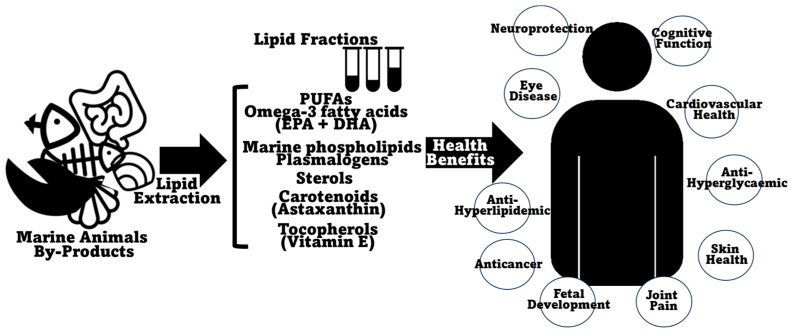
Schematic representation of the beneficial health-promoting effects and biological activities ascribed to/reported in lipids and lipid fractions of marine animal co-products.

**Figure 5 marinedrugs-22-00073-f005:**
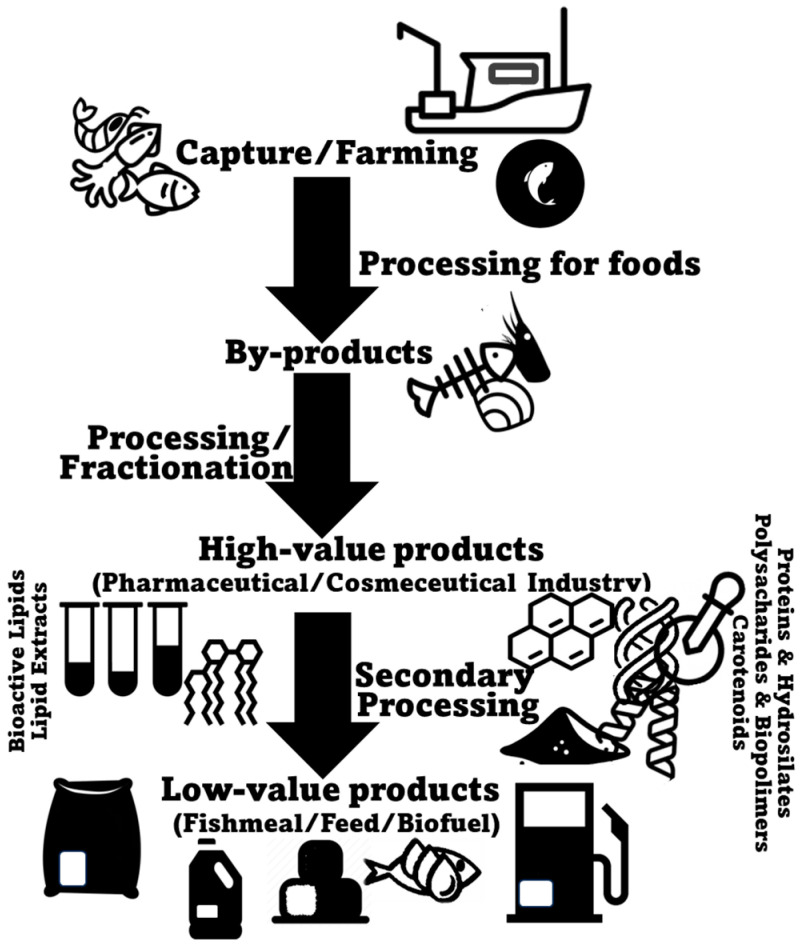
Possible framework for an integrated biorefinery approach for an exhaustive exploitation of marine animal co-products, including the extraction of valuable lipid fractions.

**Figure 6 marinedrugs-22-00073-f006:**
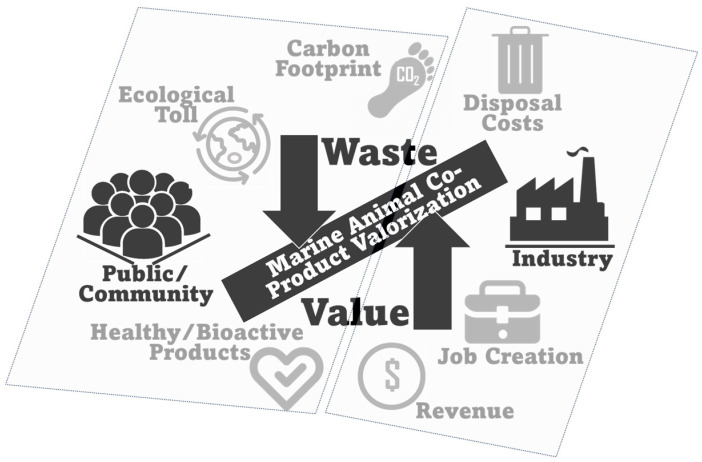
Advantages of a more rational and extensive use of lipids derived from marine animal co-products as valuable resources.

**Table 1 marinedrugs-22-00073-t001:** Studies characterizing the fatty acid profiles of fish co-products using conventional extraction methods, with an emphasis on healthy characteristics. Results are presented as percentages of total fatty acids.

Species	Co-Product	Extraction	PUFAs	Omega-3	n-6/3	EPA	DHA	Ref.
*Champsocephalus gunnari*	Brain	Solvent (hexane)	0%	0%	------	0%	0%	[110]
*Champsocephalus gunnari*	Liver	Solvent (hexane)	2.29%	1.34%	0	1.34%	0%	[110]
*Champsocephalus gunnari*	Stomach	Solvent (hexane)	35.6%	32.7%	0.02	15.9%	14.9%	[110]
*Champsocephalus gunnari*	Skin	Solvent (hexane)	27.8%	25.9%	0	15.6%	8.0%	[110]
*Clupea harengus*	Co-product mix (heads, fins, tails, and viscera)	Bligh and Dyer	35.5%	26.4%	0.34	5.6%	9.2%	[109]
*Clupea harengus*	Minced co-product (heads, frames, skin, and viscera)	Bligh and Dyer	21.9%	------	------	6.4%	9.4%	[99]
*Dicentrarchus labrax*	Heads	Folch	28.0%	12.4%	1.21	3.0%	5.1%	[93]
*Dicentrarchus labrax*	Frames	Folch	29.4%	13.1%	1.19	3.2%	5.2%	[93]
*Dicentrarchus labrax*	Skin	Folch	33.8%	16.6%	1.00	4.0%	7.5%	[93]
*Dicentrarchus labrax*	Trimmings	Folch	27.3%	11.0%	1.43	2.6%	4.3%	[93]
*Dicentrarchus labrax*	Viscera	Folch	27.9%	10.7%	1.55	2.4%	4.0%	[93]
*Euthynnus affinis*	Heads	Bligh and Dyer	28.8%	17.2%	0.67	1.5%	15.7%	[100]
*Euthynnus affinis*	Intestine	Bligh and Dyer	27.4%	17.0%	0.61	2.7%	14.3%	[100]
*Euthynnus affinis*	Liver	Bligh and Dyer	24.0%	15.9%	0.51	1.7%	14.2%	[100]
*Gadus morhua*	Offal (heads, viscera, and skeletal frames)	Bligh and Dyer	32.1%	------	------	8.9%	13.3%	[102]
*Gadus morhua*	Liver	Bligh and Dyer	24.7%	------	------	7.7%	11.4%	[102]
*Katsuwonus pelamis*	Heads	Soxhlet	12.7%	9.6%	0.32	1.3%	6.3%	[111]
*Lophius litulon*	Liver	Soxhlet	46.6%	------	------	1.2%	8.1%	[112]
*Notothenia rossii*	Brain	Solvent (hexane)	32.9%	32.2%	0.02	9.8%	22.0%	[110]
*Notothenia rossii*	Liver	Solvent (hexane)	26.0%	21.8%	0.13	8.4%	11.5%	[110]
*Notothenia* *rossii*	Stomach	Solvent (hexane)	41.6%	30.8%	0.32	11.3%	18.6%	[110]
*Notothenia rossii*	Skin	Solvent (hexane)	35.0%	31.0%	0.09	16.2%	10.7%	[110]
*Salmo salar*	Heads	Bligh and Dyer	35.4%	27.7%	0.28	8.4%	12.1%	[113]
*Salmo salar*	Heads	Folch	31.9%	16.3%	0.93	3.2%	4.8%	[93]
*Salmo salar*	Frames	Folch	31.9%	15.9%	0.98	3.0%	4.6%	[93]
*Salmo salar*	Skin	Folch	31.9%	15.4%	1.05	2.8%	4.0%	[93]
*Salmo salar*	Trimmings	Folch	32.0%	15.9%	0.98	3.0%	4.0%	[93]
*Salmo salar*	Viscera	Folch	25.0%	10.4%	1.37	1.6%	2.3%	[93]
*Sardinella lemuru*	Heads	Bligh and Dyer	26.4%	17.8%	0.54	1.8%	16.0%	[101]
*Sardinella lemuru*	Intestine	Bligh and Dyer	24.9%	13.6%	0.83	1.7%	11.9%	[101]
*Sardinella lemuru*	Liver	Bligh and Dyer	22.7%	15.7%	0.44	2.8	13.0%	[101]
*Sardinella aurita*	Viscera	Bligh and Dyer	30.5%	26.1%	0.15	7.4%	13.6%	[97]
*Sarpa salpa*	Viscera	Bligh and Dyer	34.8%	20.4%	0.71	4.1%	6.0%	[97]
*Scomber australasicus*	Head	EtOH:hexane	39.9%	36.6%	0.09	9.1%	21.9%	[104]
*Scomber australasicus*	Skin	EtOH:hexane	38.1%	34.8%	0.09	9.6%	19.5%	[104]
*Scomber australasicus*	Roe	EtOH:hexane	47.0%	44.4%	0.06	11.3%	27.5%	[104]
*Scomber australasicus*	Male gonads	EtOH:hexane	44.7%	42.5%	0.05	12.1%	24.7%	[104]
*Scomber scombrus*	Heads	Soxhlet	25.4%	------	------	3.6%	9.3%	[114]
*Scomber scombrus*	Gills	Soxhlet	12.3%	------	------	1.0%	1.7%	[114]
*Scophthalmus maximus*	Heads	Folch	36.8%	22.5%	0.61	4.4%	11.6%	[93]
*Scophthalmus maximus*	Frames	Folch	36.5%	21.7%	0.64	5.2%	7.9%	[93]
*Scophthalmus maximus*	Skin	Folch	37.4%	22.6%	0.62	5.2%	8.9%	[93]
*Scophthalmus maximus*	Trimmings	Folch	37.5%	22.8%	0.61	4.9%	9.8%	[93]
*Scophthalmus maximus*	Viscera	Folch	33.3%	17.7%	0.86	2.7%	7.6%	[93]
*Sparus aurata*	Fishbone	Bligh and Dyer	33.8%	13.6%	1.48	2.8%	4.6%	[91]
*Sparus aurata*	Frames	Folch	28.5%	12.3%	1.27	2.2%	4.8%	[93]
*Sparus aurata*	Gills	Bligh and Dyer	31.2%	11.9%	1.62	1.9%	4.1%	[91]
*Sparus aurata*	Guts	Bligh and Dyer	33.1%	12.1%	1.75	1.8%	3.5%	[91]
*Sparus aurata*	Heads	Bligh and Dyer	33.8%	14.0%	1.41	2.8%	5.0%	[91]
*Sparus aurata*	Heads	Folch	28.4%	12.7%	1.20	2.2%	5.2%	[93]
*Sparus aurata*	Liver	Bligh and Dyer	32.2%	13.6%	1.38	1.9%	4.9%	[91]
*Sparus aurata*	Skin	Bligh and Dyer	33.2%	12.9%	1.57	2.0%	4.0%	[91]
*Sparus aurata*	Skin	Folch	29.9%	13.2%	1.21	2.3%	5.5%	[93]
*Sparus aurata*	Trimmings	Folch	29.6%	13%	1.23	2.2%	5.4%	[93]
*Sparus aurata*	Viscera	Folch	28.8%	12.9%	1.20	1.7%	5.9%	[93]
*Thunnus thynnus*	Minced side streams	Folch	33.2%	29.9%	0.06	9.9%	13.6%	[96]

**Table 2 marinedrugs-22-00073-t002:** Studies characterizing the fatty acid profiles of crustacean co-products using conventional extraction methods, with an emphasis on healthy characteristics. Results are presented as percentages of total fatty acids.

Species	Co-Product	Extraction	PUFAs	Omega-3	n-6/3	EPA	DHA	Ref.
*Chionoecetes opilio*	Co-product mix (cephalothorax, digestive system, and physiological liquid)	Bligh and Dyer	24.4%	21.1%	0.10	9.9%	8.9%	[128]
Commercial crab (no specified species)	Shells	Folch	35.9%	23.2%	0.52	12.5%	9.9%	[129]
Commercial shrimp (no specified species)	Shells	Folch	40.9%	12.3%	2.2	6.3%	4.1%	[129]
*Jasus edwardsii*	Hepatopancreas	Soxhlet	7.8%	3.1%	1.52	0.9%	0.9%	[127]
*Lithodes santolla*	Exoskeleton	Bligh and Dyer	40.0%	40.0%	0	20.5%	14.4%	[55]
*Metapenaeus monoceros*	Minced co-product (heads, tails, shells)	Folch	34.5%	------	------	8.9%	6.9%	[118]
*Nephrops norvegicus*	Heads	Folch	36.2%	27.6%	0.26	15.5%	8.4%	[61]
*Pandalus borealis*	Co-product mix (heads, tails, shells)	Bligh and Dyer	43.9%	24.2%	0.57	8.9%	10.7%	[57]
*Pandalus borealis*	Processing co-product	Soxhlet	41.1%	37.1%	0.11	21.1%	13.9%	[59]
*Panulirus cygnus*	Cephalothorax	Folch	38.2%	13.5%	0.64	5.6%	4.2%	[126]
*Penaeus japonicus*	Hepatopancreas	Folch	37.2%	20.0%	0.82	8.4%	6.1%	[130]
*Penaeus kerathurus*	Cephalothorax	Bligh and Dyer	44.5%	28.7%	0.55	14.5%	13.4%	[124]
*Penaeus kerathurus*	Minced co-product (heads, tails, shells)	Folch	38.8%	------	------	12.2%	16.1%	[118]
*Penaeus monodon*	Heads	Bligh and Dyer	44.8%	29.8%	0.50	15.4%	13.3%	[123]
*Penaeus paulensis*	Minced co-product (heads, tails, shells)	Bligh and Dyer	34.6%	26.0%	0.30	11.7%	12.2%	[116]
*Penaeus vannamei*	Cephalothorax	Bligh and Dyer	43.0%	12.2%	------	5.0%	7.2%	[131]
*Penaeus vannamei*	Cephalothorax	Folch	42.5%	10.5%	------	4.1%	6.4%	[131]
*Penaeus vannamei*	Cephalothorax	Bligh and Dyer	39.3%	14.7%	1.67	4.6%	8.3%	[58]
*Penaeus vannamei*	Cephalothorax	“Typical solvent extraction”	48.5%	24.5%	0.95	9.6%	13.3%	[132]
*Penaeus vannamei*	Cephalothorax	“Solvent extraction”	37.5%	18.1%	1.08	9.2%	8.1%	[133]
*Penaeus vannamei*	Hepatopancreas	Bligh and Dyer	37.4%	10.6%	2.53	2.2%	6.2%	[58]
*Penaeus vannamei*	Hepatopancreas	Bligh and Dyer	38.1%	16.2%	1.35	3.3%	10.4%	[121]
*Pleoticus muelleri*	Shells	Bligh and Dyer	52.0%	50.3%	0.03	21.5%	22.3%	[55]
*Pleoticus muelleri*	Shells + heads	Bligh and Dyer	43.9%	42.3%	0.04	14.9%	22.0%	[55]
*Trachypena curvirostris*	Co-product mix (heads, tails, shells)	Bligh and Dyer	48.3%	26.1%	0.44	10.7%	10.9%	[57]

**Table 3 marinedrugs-22-00073-t003:** Studies characterizing the fatty acid profiles of mollusk co-products, with an emphasis on healthy characteristics. Results are presented as percentages of total fatty acids.

**Hard-Shelled Mollusks**	**Co-Product**	**Extraction**	**PUFAs**	**Omega-3**	**n-6/3**	**EPA**	**DHA**	**Ref.**
*Pecten maximus*	Pooled mantle, gill, liver, digestive gland, kidney	Supercritical extraction	42.1%	40.7%	0.03	20.0%	12.3%	[149]
*Pinna rugosa*	Viscera	Folch	41.8%	37.5%	------	17.0%	20.0%	[119]
**Cephalopods**	**Co-product**	**Extraction**	**PUFAs**	**Omega-3**	**n-6/3**	**EPA**	**DHA**	**Ref.**
Commercial squid (no specified species)	Squid viscera oil	Supercritical extraction	44.7%	------	------	15.1%	24.9%	[69]
*Doryteuthis gahi*	Pooled viscera, heads, skin	Bligh and Dyer	52.6%	48.6%	0.08	17.2%	30.8%	[150]
*Dosidicus gigas*	Viscera	Folch	36.6%	34.0%	------	15.5%	17.8%	[119]
*Illex argentinus*	Viscera	Wet pressing	------	------	0.75	9.3%	16.4%	[151]
*Octopus vulgaris*	Pooled co-products	Bligh and Dyer	49.3%	36.8%	0.34	12.9%	22.2%	[67]
*Sepia officinalis*	Viscera	Bligh and Dyer	44.0%	26.0%	0.69	11.6%	6.3%	[97]

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
