# Peer review of "Marine Animal Co-Products—How Improving Their Use as Rich Sources of Health-Promoting Lipids Can Foster Sustainability"

_marinedrugs, 2024, doi:10.3390/md22020073_

Round 1

Reviewer 1 Report

Comments and Suggestions for Authors

This is a good review concerning the current status of recovery & use of marine animal co-products. The paper first describes the types and size of waste materials produced by commerciallization of fish, crustaceans and molluscs, and their corresponding contents on healthy lipids, followed by a description of the Value of Marine Animal Co-Product Lipids for Human Health, and their current applications in the industry. Next, the Sustainability and Environmental Impact of annual discards from the world fisheries is described together with the need of recycling such large amounts of biomass as a challenging task for the close future. The extraction and valorization of lipids from marine animal co-product matrixes is envisioned as a necessary task, where some of the most relevant challenges underlying a sustainable and profitable exploration of lipids are described.

This manuscript describes how lipid sources remain significantly undervalued and the potential  of their full valorization that may generate both financial and ecological benefits.

The potential use of marine animal co-products as sustainable sources of health promoting and bioactive lipids is nicely presented on the basis of a detailed analyses of 306 literature refs

The paper is well structured and carefully writen. It is an interesting review that deserves its publication. Only some few typographycal errors must be corrected, as follows:

Page 2 Line 96 being a review made at the end of 2023 a more recent estimation for  must be given for tons of aquatic animals production.

Page 3 L 109 the Word “lipdiomics” must be corrected

Page 6 L 211 correct the Word “uising”

Page 12 L424 correct the Word “benefitial”

Page 17 L 650, correct the phrase “hexanic extract of was shown…”

Page 19 L 759- remove one of the two “in” :dying agent in in animal feed…

Page 22 L 853- add a second ¨t” to “t to beter understand…”

Author Response

This is a good review concerning the current status of recovery & use of marine animal co-products. The paper first describes the types and size of waste materials produced by commerciallization of fish, crustaceans and molluscs, and their corresponding contents on healthy lipids, followed by a description of the Value of Marine Animal Co-Product Lipids for Human Health, and their current applications in the industry. Next, the Sustainability and Environmental Impact of annual discards from the world fisheries is described together with the need of recycling such large amounts of biomass as a challenging task for the close future. The extraction and valorization of lipids from marine animal co-product matrixes is envisioned as a necessary task, where some of the most relevant challenges underlying a sustainable and profitable exploration of lipids are described.

This manuscript describes how lipid sources remain significantly undervalued and the potential  of their full valorization that may generate both financial and ecological benefits.

The potential use of marine animal co-products as sustainable sources of health promoting and bioactive lipids is nicely presented on the basis of a detailed analyses of 306 literature refs

The paper is well structured and carefully writen. It is an interesting review that deserves its publication. Only some few typographycal errors must be corrected, as follows:

Page 2 Line 96 being a review made at the end of 2023 a more recent estimation for  must be given for tons of aquatic animals production.

Reply: We thank the reviewer for the kind comments and the suggestions to improve our manuscript. Since the last FAO report is dated from 2022 and only contains data until 2020, it is really hard to find reliable information about aquatic animal production after that. We did find an update regarding the production and capture of fish in specific, which we included now in the manuscript: “In the specific case of fish, the aquaculture production (94.7 million metric tons) was expected to have slightly surpass the production by capture fisheries (90.7 million metric tons) in 2023 [1].“

Page 3 L 109 the Word “lipdiomics” must be corrected

Reply: The word was corrected accordingly.

Page 6 L 211 correct the Word “uising”

Reply: The word was corrected accordingly.

Page 12 L424 correct the Word “benefitial”

Reply: The word was corrected accordingly.

Page 17 L 650, correct the phrase “hexanic extract of was shown…”

Reply: The phrase was corrected as suggested.

Page 19 L 759- remove one of the two “in” :dying agent in in animal feed…

Reply: The extra “in” was removed from the manuscript.

Page 22 L 853- add a second ¨t” to “t to beter understand…”

Reply: The correction was made accordingly.

References:

  1. OECD/FAO, OECD-FAO Agricultural Outlook. 2023.

Reviewer 2 Report

Comments and Suggestions for Authors

Review: marinedrugs-2790070

Title: Marine Animal Co-Products – How Improving Their Use as Rich Sources of Health-Promoting Lipids Can Foster Sustainability

Summary:

The authors describe the potential for marine animal co-products as a valuable resource for obtaining health-promoting lipids, including omega-3 polyunsaturated fatty acids. Reducing waste in the seafood industry by harnessing these underutilized co-products for use in consumer products is also suggested for enhancing sustainability. The manuscript is well written and outlines the potential value of the various co-products, suggesting opportunities for more eco-conscious practices. To improve the significance of this work and provide a comprehensive review, additional text should be added to highlight regional differences in the use of biproducts and detail the economic value of low-end vs high-end repurposing of these resources. Other suggested revisions are detailed below.

Revisions:

1.     Portions of this review read more like an opinion paper than a review. The text should be revised to demonstrate the reported consensus more accurately on the topics discussed as it relates to health-promoting lipids, feasibility of co-products for the various uses suggested and impact on sustainability. Summarizing and referencing progress that has been made globally to use marine animal co-products would be helpful to the reader.

2.      This review lacks a discussion of geographical differences in the use of marine animal co-products as seems to focus primarily on global trends. A discussion of the differences in co-product usage between regions (continents/countries/demographics) should be added. Importantly, are there major differences in the amount of waste being generated based on location?

3.      A discussion of the specific economic potential, referencing appropriate sources for this data, should be added. Are there estimates for the value of these co-products or data on revenue generated in countries that are utilizing more of these animal co-products?

4.      Add a discussion of the known/approved pharmaceutical uses for these co-products worldwide. References demonstrating the utilization of these co-products for higher-end purposes would be beneficial to readers.

5.      Nearly all the figures are oversimplified and do not enhance the readers’ understanding of the text. Some of the figures are also misleading, such as figure 4, which is not sufficiently supported by the references. The authors should avoid health-claims that are not widely reported and accepted for these co-products. Many of the referenced biological activities and health benefits are not substantiated in this review. All figures should be revised to add value to this review.

6.      Although the authors mention some of the potential challenges of using marine animal co-products for human consumption/use, there is a clear omission of marine toxins and impact of harmful algal blooms on these resources. The text also reads in a way that plays down the potential issues with contamination of these co-products such as the viscera where toxins often accumulate. The language should be corrected to define the known issues more clearly regarding contaminants, especially the bioaccumulation of many toxicants and toxins in these tissues.

7.      Minor edits to spelling and grammar are needed. Abbreviations such as HACCP should also be defined first.

Comments on the Quality of English Language

English quality is acceptable. Only a few minor typos were detected. 

Author Response

Review: marinedrugs-2790070

Title: Marine Animal Co-Products – How Improving Their Use as Rich Sources of Health-Promoting Lipids Can Foster Sustainability 

Summary:

The authors describe the potential for marine animal co-products as a valuable resource for obtaining health-promoting lipids, including omega-3 polyunsaturated fatty acids. Reducing waste in the seafood industry by harnessing these underutilized co-products for use in consumer products is also suggested for enhancing sustainability. The manuscript is well written and outlines the potential value of the various co-products, suggesting opportunities for more eco-conscious practices. To improve the significance of this work and provide a comprehensive review, additional text should be added to highlight regional differences in the use of biproducts and detail the economic value of low-end vs high-end repurposing of these resources. Other suggested revisions are detailed below.

Revisions:

  1.    Portions of this review read more like an opinion paper than a review. The text should be revised to demonstrate the reported consensus more accurately on the topics discussed as it relates to health-promoting lipids, feasibility of co-products for the various uses suggested and impact on sustainability. Summarizing and referencing progress that has been made globally to use marine animal co-products would be helpful to the reader.

Reply: We thank the reviewer for the comments. When you have sections of the manuscripts that deal with “Challenges and Future Directions”, or even “Sustainability and Environmental Impact”, which are necessarily written with an eye on the future, these kind of imply a personal view or conception of the future of the use of these co-products. We believe these are important sections in the paper, if the objective is to properly characterize the state of the use of marine animal co-products for the purpose of obtaining healthy lipids, and set the stage and vindicate a more rational use of these resources in the future. However, this is a paper with more than 300 references and an effort has been made to substantiate every claim with references throughout all sections. Therefore, taking into account the effort made to gather and review all the literature presented, this should still be considered essentially a review paper.

On the subject of demonstrating the current consensus related to health promoting lipids, we included additional information in the manuscript in beginning of the “4. The Value of Marine Animal Co-Product Lipids for Human Health” section, which already alluded to this subject:

“Lipids make up a wide group of essential macronutrients which are incorporated through the diet, which represent energy resources for the cell while also performing other roles in important physiological functions such as cellular signaling and regulation of cell membrane physical characteristics and function [1]. Unbalanced lipid intake can cause either a deficiency in essential fatty acids and fat-soluble vitamins [2-4], or in excess, can lead to problems linked with the metabolic syndrome, like hypertension, diabetes and cardiovascular but also liver disease [5, 6]. Recent knowledge has pointed out diacylglycerols and structured (and PUFA-rich) triglycerides and phospholipids (especially omega-3-containing phospholipids) as representing improved nutritional and health value [7].”

Information about other current matters that are (mostly) consensual regarding the health promoting characteristics of lipids, such as the importance of dietary PUFA, the increasing relevance given to omega-3 fatty acids, and the important characteristics phospholipids as convenient vectors for omega-3 delivery are already discussed in the manuscript (Section “4. The Value of Marine Animal Co-Product Lipids for Human Health”).

Regarding the feasibility of using co-products for the various uses suggested, additional information has also been included for each case in the “5. The Value of Marine Animal Co-Product Lipids for Industry” section. In the first case, suggesting the exploitation of these resources as sources of omega-3 fatty acids for incorporation in food products, we have added the following “The use of omega-3 fatty acids derived from marine animal co-products should be enticing given the significant general occurrence of these fatty acids in these resources, and should be currently only limited by further prospection and characterization efforts and regulatory restrictions. However, there are already examples of incorporation from oils from marine animal co-products in foods, such as bakery, dairy and meat products [8].”

Regarding the potential use of omega-3 fatty acids from marine animal co-products in nutritional supplements, the text was reformulated and complementary information was added: “Omega-3 supplements are commercially available in a variety of different formulations from different brands, mostly as (fish) oils or as capsules, thought to promote an increase shelf time absorption [9-12]. In alternative to the direct commercialization of marine oils, formulations (concentrates or nanoliposome-containing) of EPA and DHA specifically can also be used as supplements in human nutrition [13, 14].” and “Therefore, the market actually exists for additional healthy lipid sources, and the use of omega-3 fatty acids derived from marine animal co-products should only depend on further technical advances for the optimization of the extraction yields using green approaches and regulatory adjustments.”

Regarding the possible use of marine animal co-products as sources of marine phospholipids with upside for the pharmaceutical industry, the following has been added: “The exploration of marine animal co-products as viable and sustainable sources of marine phospholipids with upside for the pharmaceutical industry is currently unexplored. In fact, only a limited number of reports exist characterizing and quantifying marine phospholipids in marine animal co-products. Further lipidomic characterization of these matrixes, along with the development of green strategies to isolate these compounds, will help to signal preferentially rich sources and determine the viability of exploring these resources for marine phospholipid isolation.”

When discussing the feasibility of using marine animal co-products as putative sources of plasmalogens, the following text has been added: “Currently, a possible significant use of marine animal co-products as sources of plasmalogens with interest for the pharmaceutical industry is mostly hindered by the lack of available information and proper in-depth lipidomic characterization of the content of these resources at a molecular level.”

Finally, with what regards the feasibility of using lipids from marine animal co-products in the cosmeceutical industry, the following has been added: “Moreover, fish oils, some of which produced using seafood industry co-products, have been ascribed broad benefits for the maintenance of skin homeostasis as well as in skin disorder contexts, including  photoaging, cutaneous carcinogenesis, dermatitis, cutaneous wounds and hyperpigmentation [15]. The skin health-promoting characteristics of these oils are generally related to their content in omega-3 fatty acids, especially EPA and DHA [15]. This evidence should justify further interest in studying the potential and activities of lipid fractions from marine animal co-products in skin care, and therefore uncover additional value for the cosmeceutical industry.”

Concerning the importance of lipids of marine animal co-product origin in the context of aquaculture, the feasibility of the use of these resources is already well documented in the text, including specific examples.

Regarding the issue of the current consensus about the impact of the use of marine animal co-products in sustainability and progress that was made in the use of marine animal co-products, additional information was added near the beginning of the “6. Sustainability and Environmental Impact” section: “The exploitation of the use of marine animal co-products in a “Waste to Wealth” approach, representing economic, environmental and food security upside, has been widely explored [16]. However, the strategic management of marine animal co-products must further employ the concepts of circular economy and life cycle thinking in order to increase their efficient use and mitigate the environmental impact of the seafood industry [17]. Marine animal co-products are generally recognized for having an interesting composition from a chemical standpoint, including valuable protein and lipid fractions, minerals, enzymes and vitamins [12]. Currently, there are several main paths established for the generation of value from marine animal co-products: the obtention of marine proteins (fishmeal, silage and hydrolysates), the production of PUFA-enriched oils, the production of biodiesel and biogas, and the isolation of higher end compounds such as vitamins, enzymes, minerals, taurine and creatine and hydroxyapatite directed for specific industrial or pharmaceutical uses [18]. A more efficient and profitable use of marine animal co-products depends on new valorization approaches and improved recovery technologies for the already signaled and novel compounds of interest detected [19, 20], and in this case, the specific presence of bioactive lipids has been patently disregarded.”

  1. This review lacks a discussion of geographical differences in the use of marine animal co-products as seems to focus primarily on global trends. A discussion of the differences in co-product usage between regions (continents/countries/demographics) should be added. Importantly, are there major differences in the amount of waste being generated based on location?

Unfortunately, like we state in the manuscript at some point there is not much information available about the generation of marine animal co-products: “When writing the present review, other than estimates, it was challenging to find credible information on the true numbers of marine animal co-product production at global level.” The same lack of information also occurs at local level, with just minimal reliable information being made available on the subject by authorities. Nevertheless, we found some information at the IFFO - The Marine Ingredients Organization site that we are including, along with other trends that may be relevant for this subject. We added this discussion at the end of the “2. Marine Animal Co-Products” section: “Taking into account geographical idiosyncratic tendencies regarding both the production/capture and consumption trends driven by regional socioeconomic and cultural constraints, differences in the production of co-products both regarding quantity and types are to be expected. According to the IFFO - The Marine Ingredients Organization site, Asia is the main responsible for the recycling of raw materials from the seafood industry, accounting for the production of 40.0% of marine ingredients based on co-products, followed by Europe (23.3%) and Latin America (20.8%), with North America producing just 7.8% of global co-products [21]. This is somewhat expected since in 2020, Asian countries distinctly lead total fishery production (70%) [22]. Although considerable efforts have been produced to more readily utilize marine animal co-products originated from the seafood processing worldwide, their exploration and use may vary considerably regionally [23]. In Asia, seafood value chains effectively accommodate these resources resulting in little waste, while in Europe, a stricter legislation (as response to bovine spongiform encephalopathy and other food threats) has rendered the utilization of these resources more difficult [24, 25]. In other regions less developed, a more lenient legislation and the low valorization of the co-products make disposal more generalized [23]. As specific cases of a specially efficient utilization of marine animal co-products we have the case of Norway, which has developed processing facilities able to process over 0.65 million tons of seafood co-products by year, and where the Norwegian Atlantic salmon industry is reported to utilize 90% of its byproducts [26].”

  1. A discussion of the specific economic potential, referencing appropriate sources for this data, should be added. Are there estimates for the value of these co-products or data on revenue generated in countries that are utilizing more of these animal co-products?

Again, there not many sources available with reliable information or projections on this subject. We include the information presented on the Future Market Insights (FMI) site. We also make mention of the case of Scotland, where studies have been conducted presenting concrete evidence of economic upside in the use of marine animal co-products. This information was added to the “1. Introduction” section: “The global marine co-products market was proposed to represent 33.7 million dollars in 2023, with the current projections anticipating this value to almost double by 2033 (64.8 million dollars), with this growth being proposed to be essentially driven by the expansion of the mariculture sector, technical advances in extraction and processing techniques and the development of  novel applications as functional foods and nutraceuticals [27]. The importance of a rational and holistic perspective for the repurposing of marine animal co-products may be essential for the generation of revenue. A study appraising the effectiveness of the Scottish salmon farming system proposed that, although resources are generally well utilized, the co-product value output could be improved by 803%, representing 5.5% more value to the salmon industry [16]. This could be achieved by a strategic management of the co-products (heads, frames, trimmings and belly flaps) focusing on optimizing edible yield and repurposing and direct them specifically for domestic and foreign food markets [16]. In fact, while salmon co-products may not especially appreciated in some markets, they may be in high demand in others, and that is the case of salmon heads in Vietnam, frames in Eastern Europe, and belly flaps in Japan [28].”

The economic upside of the use of marine animal co-products in the context of the fish oil market and as sources of omega-3 for the food/supplement industries are already evidenced in the manuscript:

“The fish oil market represented a total of 1905.77 million dollars in 2019 and is continually growing [29], with fish oil produced from fish co-products representing 26% of total production in 2016 [30].”

“The global lipid nutrition market, mostly based on the promise of omega-3 fatty acids as health promoters, is expected to reach 17 million dollars by 2031, growing by 7.6% annually over the 2021-2031 period, driven by increasing demand [31].”

  1. Add a discussion of the known/approved pharmaceutical uses for these co-products worldwide. References demonstrating the utilization of these co-products for higher-end purposes would be beneficial to readers.

Reply: A discussion about implemented uses of compounds derived from marine animal co-products for higher end purposes (pharmaceutical and cosmeceutical industries) was added to the section “5. The Value of Marine Animal Co-Product Lipids for Industry”:

“There are several compounds that are extracted from marine animal co-products with established demand for high-end uses in the pharmaceutical and cosmeceutical industries. Collagen, gelatin, and collagen derivatives [32, 33], protein hydrolysates [32], chitin and chitosan [32-37], glycosaminoglycans [12, 38] and hydroxyapatite [39, 40] in particular have well established importance in the pharmaceutical, cosmeceutical and biomedicine industries (in addition to the aforementioned astaxanthin). The prospection of bioactive lipids and the investigation of the biological activities of lipid extracts from marine animal co-products have the potential to add to this list and to increase the value of these resources and expanding their applications for higher-end purposes.”

Please note that, for now and to the authors best knowledge, there are no concrete applications of particular lipid species or lipid extracts obtained from marine animal co-products specifically in such industries. In the manuscript, we do highlight the cases of astaxanthin and tocopherols that occur plentifully in some cases of marine animal co-products and should be present in lipid fractions of such resources and acknowledge their importance specifically in the case of the cosmeceutical industry.

  1. Nearly all the figures are oversimplified and do not enhance the readers’ understanding of the text. Some of the figures are also misleading, such as figure 4, which is not sufficiently supported by the references. The authors should avoid health-claims that are not widely reported and accepted for these co-products. Many of the referenced biological activities and health benefits are not substantiated in this review. All figures should be revised to add value to this review.

Reply: An effort was made to make all figures more informative, and Figure 4 in particular was modified in order to include only the most consensual health benefits of lipids known to occur in marine co-products.

  1. Although the authors mention some of the potential challenges of using marine animal co-products for human consumption/use, there is a clear omission of marine toxins and impact of harmful algal blooms on these resources. The text also reads in a way that plays down the potential issues with contamination of these co-products such as the viscera where toxins often accumulate. The language should be corrected to define the known issues more clearly regarding contaminants, especially the bioaccumulation of many toxicants and toxins in these tissues.

Reply: A discussion about the problem that the presence of marine toxins and other toxicants may represent for the use of marine animal co-products was included in the “7. Challenges and Future Directions” section, where problems related with the presence of contaminants (metals (and metalloids), pesticides, polychlorinated biphenyls, dioxins and (micro)plastics) was already mentioned: “Moreover, another technical challenge may have to do with the potential presence of contaminants in marine animal co-products, namely those known to accumulate in marine animal biomass, such as metals (and metalloids), pesticides, polychlorinated biphenyls, dioxins and (micro)plastics [126, 284, 326-328]. Contaminant accumulation has been reported in crustacean hepatopancreas [41-43], and in the skin, viscera, bones and scales of fish [44-48]. Mollusks also accumulate toxicants [49-52], with the digestive glands and gills of hard-shelled animals representing preferential accumulation sites  [53-55]. The presence of marine biotoxins may represent a very concrete challenge for the use of marine animal co-products. These biotoxins, produced by bacteria, cyanobacteria, and microalgae, are known to bioaccumulate in fish, mollusks and crustaceans [56-60]. In the case of the presence of biotoxins in fish co-products in particular, this may in fact represent a problem since fish appear to preferentially accumulate toxins in the viscera rather than in the flesh [61-65]. Therefore, it becomes paramount to signal the collection of fish and marine animals in the vicinities of harmful algal blooms incidence where these toxins are abundantly produced [66], even for purposes of making use of animal parts not directly intended for dietary consumption. These concerns about the presence of contaminants in marine animal co-products should not be taken lightly and may imply additional thorough quality control measures and investments in hazard analysis and critical control point (HACCP) and decontamination procedures, ensuring the safety and purity of lipid extracts.”

  1. Minor edits to spelling and grammar are needed. Abbreviations such as HACCP should also be defined first.

Reply: The text was thoroughly reviewed for spelling and grammar errors, as also suggested by other reviewers. “Hazard analysis and critical control point” (HACCP) is now properly defined in the text.

References:

  1. Ghazani, S. M.; Marangoni, A. G., Microbial lipids for foods. Trends in Food Science & Technology 2022, 119, 593-607.
  2. Shahidi, F.; Pinaffi-Langley, A. C. C.; Fuentes, J.; Speisky, H.; de Camargo, A. C., Vitamin E as an essential micronutrient for human health: Common, novel, and unexplored dietary sources. Free Radical Biology and Medicine 2021, 176, 312-321.
  3. Gimenez, M. S.; Oliveros, L. B.; Gomez, N. N., Nutritional deficiencies and phospholipid metabolism. International Journal of Molecular Sciences 2011, 12, (4), 2408-33.
  4. Jeppesen, P.; Christensen, M.; Høy, C.; Brøbech, P., Essential fatty acid deficiency in patients with severe fat malabsoption. The American Journal of Clinical Nutrition 1997, 65, 837-43.
  5. Kuller, L. H., Dietary fat and chronic diseases: epidemiologic overview. Journal of the American Dietetic Association 1997, 97, (7 Suppl), S9-15.
  6. Graham, D. S.; Liu, G.; Arasteh, A.; Yin, X. M.; Yan, S., Ability of high fat diet to induce liver pathology correlates with the level of linoleic acid and Vitamin E in the diet. PLoS One 2023, 18, (6), e0286726.
  7. Xu, Q.; Tang, Q.; Xu, Y.; Wu, J.; Mao, X.; Li, F.; Wang, S.; Wang, Y., Biotechnology in Future Food Lipids: Opportunities and Challenges. Annual Review of Food Science and Technology 2023, 14, 225-246.
  8. Pateiro, M.; Domínguez, R.; Varzakas, T.; Munekata, P. E. S.; Movilla Fierro, E.; Lorenzo, J. M., Omega-3-Rich Oils from Marine Side Streams and Their Potential Application in Food. Marine Drugs 2021, 19, (5).
  9. Venugopalan, V. K.; Gopakumar, L. R.; Kumaran, A. K.; Chatterjee, N. S.; Soman, V.; Peeralil, S.; Mathew, S.; McClements, D. J.; Nagarajarao, R. C., Encapsulation and Protection of Omega-3-Rich Fish Oils Using Food-Grade Delivery Systems. Foods 2021, 10, (7), 1566.
  10. Srigley, C.; Rader, J. I. T., Content and Composition of Fatty Acids in Marine Oil Omega-3 Supplements. Journal of Agricultural and Food Chemistry 2014, 62, (29), 7268-7278.
  11. Erkan, O. N.; Tunçelli, İ. C. A. N.; Özden, Ö., Content and economic evaluation of omega-3 fatty acid nutritional supplements. Journal of Food and Nutrition Research 2023, 62, (1), 14-25.
  12. Shahidi, F.; Varatharajan, V.; Peng, H.; Senadheera, R., Utilization of marine by-products for the recovery of value-added products. Journal of Food Bioactives 2019, 6, (0).
  13. Bimbo, A. P., Raw material sources for the long-chain omega-3 market: Trends and sustainability. Part 2. . 99th AOCS Annual Meeting and Expo in Seattle 2009.
  14. Rasti, B.; Erfanian, A.; Selamat, J., Novel nanoliposomal encapsulated omega-3 fatty acids and their applications in food. Food Chemistry 2017, 230, 690-696.
  15. Huang, T. H.; Wang, P. W.; Yang, S. C.; Chou, W. L.; Fang, J. Y., Cosmetic and Therapeutic Applications of Fish Oil's Fatty Acids on the Skin. Marine Drugs 2018, 16, (8).
  16. Stevens, J. R.; Newton, R. W.; Tlusty, M.; Little, D. C., The rise of aquaculture by-products: Increasing food production, value, and sustainability through strategic utilisation. Marine Policy 2018, 90, 115-124.
  17. Cooney, R.; de Sousa, D. B.; Fernández-Ríos, A.; Mellett, S.; Rowan, N.; Morse, A. P.; Hayes, M.; Laso, J.; Regueiro, L.; Wan, A. H. L.; Clifford, E., A circular economy framework for seafood waste valorisation to meet challenges and opportunities for intensive production and sustainability. Journal of Cleaner Production 2023, 392, 136283.
  18. Kaanane, A.; Mkadem, H., Valorization Technologies of Marine By-Products. In Innovation in the Food Sector Through the Valorization of Food and Agro-Food By-Products, Ana Novo de, B.; Irene, G., Eds. IntechOpen: Rijeka, 2020; p Ch. 3.
  19. de la Caba, K.; Guerrero, P.; Trung, T. S.; Cruz-Romero, M.; Kerry, J. P.; Fluhr, J.; Maurer, M.; Kruijssen, F.; Albalat, A.; Bunting, S.; Burt, S.; Little, D.; Newton, R., From seafood waste to active seafood packaging: An emerging opportunity of the circular economy. Journal of Cleaner Production 2019, 208, 86-98.
  20. Venugopal, V., Valorization of Seafood Processing Discards: Bioconversion and Bio-Refinery Approaches. Frontiers in Sustainable Food Systems 2021, 5.
  21. Organisation, I.-T. M. I. The global growth of by-products. https://www.iffo.com/global-growth-products (2024),
  22. Zhang, J.; Akyol, Ç.; Meers, E., Nutrient recovery and recycling from fishery waste and by-products. Journal of Environmental Management 2023, 348, 119266.
  23. Pounds, A.; Kaminski, A. M.; Budhathoki, M.; Gudbrandsen, O.; Kok, B.; Horn, S.; Malcorps, W.; Mamun, A. A.; McGoohan, A.; Newton, R.; Ozretich, R.; Little, D. C., More Than Fish-Framing Aquatic Animals within Sustainable Food Systems. Foods 2022, 11, (10).
  24. Regueiro, L.; Newton, R.; Soula, M.; Méndez, D.; Kok, B.; Little, D. C.; Pastres, R.; Johansen, J.; Ferreira, M., Opportunities and limitations for the introduction of circular economy principles in EU aquaculture based on the regulatory framework. Journal of Industrial Ecology 2022, 26, (6), 2033-2044.
  25. Woodgate, S. L.; Wilkinson, R. G., The role of rendering in relation to the bovine spongiform encephalopathy epidemic, the development of EU animal by-product legislation and the reintroduction of rendered products into animal feeds. Annals of Applied Biology 2021, 178, (3), 430-441.
  26. Kumar, V.; Muzaddadi, A.; Mann, S.; Balakrishnan, R.; Bembem, K.; Kalnar, Y., Utilization of Fish Processing Waste: A Waste to Wealth Approach. 2022; pp 127-131.
  27. (FMI), F. M. I. Marine By-products Market. https://www.futuremarketinsights.com/reports/marine-by-products-market (2024),
  28. Shahidi, F., Maximising the value of marine by-products. Woodhead Publishing: 2006.
  29. Coppola, D.; Lauritano, C.; Palma Esposito, F.; Riccio, G.; Rizzo, C.; de Pascale, D., Fish Waste: From Problem to Valuable Resource. Marine Drugs 2021, 19, (2).
  30. Thirukumaran, R.; Anu Priya, V. K.; Krishnamoorthy, S.; Ramakrishnan, P.; Moses, J. A.; Anandharamakrishnan, C., Resource recovery from fish waste: Prospects and the usage of intensified extraction technologies. Chemosphere 2022, 299, 134361.
  31. yahoo!finance Global Lipid Nutrition Market Report to 2031 - by Product, Source, Form, Application, Distribution and Region. https://finance.yahoo.com/news/global-lipid-nutrition-market-report-102800783.html
  32. Espinales, C.; Romero-Peña, M.; Calderón, G.; Vergara, K.; Cáceres, P. J.; Castillo, P., Collagen, protein hydrolysates and chitin from by-products of fish and shellfish: An overview. Heliyon 2023, 9, (4), e14937.
  33. Siahaan, E. A.; Agusman; Pangestuti, R.; Shin, K. H.; Kim, S. K., Potential Cosmetic Active Ingredients Derived from Marine By-Products. Marine Drugs 2022, 20, (12).
  34. Rinaudo, M., Chitin and chitosan: Properties and applications. Progress in Polymer Science 2006, 31, (7), 603-632.
  35. Morganti, P.; Morganti, G.; Morganti, A., Transforming nanostructured chitin from crustacean waste into beneficial health products: a must for our society. Nanotechnology, Science and Applications 2011, 4, 123-9.
  36. Massironi, A.; Morelli, A.; Puppi, D.; Chiellini, F., Renewable Polysaccharides Micro/Nanostructures for Food and Cosmetic Applications. Molecules 2020, 25, (21), 4886.
  37. Casadidio, C.; Peregrina, D. V.; Gigliobianco, M. R.; Deng, S.; Censi, R.; Di Martino, P., Chitin and Chitosans: Characteristics, Eco-Friendly Processes, and Applications in Cosmetic Science. Marine Drugs 2019, 17, (6).
  38. Abdallah, M. M.; Fernández, N.; Matias, A. A.; Bronze, M. d. R., Hyaluronic acid and Chondroitin sulfate from marine and terrestrial sources: Extraction and purification methods. Carbohydrate Polymers 2020, 243, 116441.
  39. Hernández-Ruiz, K. L.; López-Cervantes, J.; Sánchez-Machado, D. I.; Martínez-Macias, M. d. R.; Correa-Murrieta, M. A.; Sanches-Silva, A., Hydroxyapatite recovery from fish byproducts for biomedical applications. Sustainable Chemistry and Pharmacy 2022, 28, 100726.
  40. Anil, S.; Sweety, V. K.; Joseph, B., Marine-Derived Hydroxyapatite for Tissue Engineering Strategies. In Handbook of the Extracellular Matrix: Biologically-Derived Materials, Maia, F. R. A.; Oliveira, J. M.; Reis, R. L., Eds. Springer International Publishing: Cham, 2023; pp 1-26.
  41. Yu, Y.; Hu, L.; Tian, D.; Yu, Y.; Lu, L.; Zhang, J.; Huang, X.; Yan, M.; Chen, L.; Wu, Z.; Shi, W.; Liu, G., Toxicities of polystyrene microplastics (MPs) and hexabromocyclododecane (HBCD), alone or in combination, to the hepatopancreas of the whiteleg shrimp, Litopenaeus vannamei. Environmental Pollution 2023, 329, 121646.
  42. Ariano, A.; Scivicco, M.; D'Ambola, M.; Velotto, S.; Andreini, R.; Bertini, S.; Zaccaroni, A.; Severino, L., Heavy Metals in the Muscle and Hepatopancreas of Red Swamp Crayfish (Procambarus clarkii) in Campania (Italy). Animals (Basel) 2021, 11, (7).
  43. Bodin, N.; Abarnou, A.; Le Guellec, A. M.; Loizeau, V.; Philippon, X., Organochlorinated contaminants in decapod crustaceans from the coasts of Brittany and Normandy (France). Chemosphere 2007, 67, (9), S36-S47.
  44. Lee, S.-J.; Mamun, M.; Atique, U.; An, K.-G., Fish Tissue Contamination with Organic Pollutants and Heavy Metals: Link between Land Use and Ecological Health. Water 2023, 15, (10), 1845.
  45. Rahman, S. A.; Abdullah, N. A.; Chowdhury, A. J. K.; Yunus, K., Fish Scales as a Bioindicator of Potential Marine Pollutants and Carcinogens in Asian Sea Bass and Red Tilapia within the Coastal Waters of Pahang, Malaysia. Journal of Coastal Research 2018, 82, (sp1), 120-125, 6.
  46. Mendoza, L. C.; Nolos, R. C.; Villaflores, O. B.; Apostol, E. M. D.; Senoro, D. B., Detection of Heavy Metals, Their Distribution in Tilapia spp., and Health Risks Assessment. Toxics 2023, 11, (3).
  47. Cordeli , A. N.; Oprea, L.; Crețu, M.; Dediu, L.; Coadă, M. T.; Mînzală, D.-N., Bioaccumulation of Metals in Some Fish Species from the Romanian Danube River: A Review. Fishes 2023, 8, (8), 387.
  48. Staniskiene, B. M., P. Budreckiene, R. Skibniewska, K. A., Distribution of Heavy Metals in Tissues of Freshwater Fish in Lithuania. Polish Journal of Environmental Studies 2006, 15, (4), 585-591.
  49. Chen, L.; Cai, X.; Cao, M.; Liu, H.; Liang, Y.; Hu, L.; Yin, Y.; Li, Y.; Shi, J., Long-term investigation of heavy metal variations in mollusks along the Chinese Bohai Sea. Ecotoxicology and Environmental Safety 2022, 236, 113443.
  50. Wang, R.; Mou, H.; Lin, X.; Zhu, H.; Li, B.; Wang, J.; Junaid, M.; Wang, J., Microplastics in Mollusks: Research Progress, Current Contamination Status, Analysis Approaches, and Future Perspectives. Frontiers in Marine Science 2021, 8.
  51. Guangyuan, L., Heavy Metals in Bivalve Mollusks. 2017; pp 553-594.
  52. Barchiesi, F.; Branciari, R.; Latini, M.; Roila, R.; Lediani, G.; Filippini, G.; Scortichini, G.; Piersanti, A.; Rocchegiani, E.; Ranucci, D., Heavy Metals Contamination in Shellfish: Benefit-Risk Evaluation in Central Italy. Foods 2020, 9, (11).
  53. Green, D. S.; Colgan, T. J.; Thompson, R. C.; Carolan, J. C., Exposure to microplastics reduces attachment strength and alters the haemolymph proteome of blue mussels (Mytilus edulis). Environmental Pollution 2019, 246, 423-434.
  54. Liu, M.; Fan, S.; Rong, Z.; Qiu, H.; Yan, S.; Ni, H.; Dong, Z., Exposure to polychlorinated biphenyls (PCBs) affects the histology and antioxidant capability of the clam Cyclina sinensis. Frontiers in Marine Science 2023, 10.
  55. Pizzurro, F.; Nerone, E.; Ancora, M.; Di Domenico, M.; Mincarelli, L. F.; Cammà, C.; Salini, R.; Di Renzo, L.; Di Giacinto, F.; Corbau, C.; Bokan, I.; Ferri, N.; Recchi, S., Exposure of Mytilus galloprovincialis to Microplastics: Accumulation, Depuration and Evaluation of the Expression Levels of a Selection of Molecular Biomarkers. Animals 2024, 14, (1), 4.
  56. Mafra, L. L.; de Souza, D. A.; Menezes, M.; Schramm, M. A.; Hoff, R., Marine biotoxins: latest advances and challenges toward seafood safety, using Brazil as a case study. Current Opinion in Food Science 2023, 53, 101078.
  57. Brett, M. M., Food poisoning associated with biotoxins in fish and shellfish. Current Opinion in Infectious Diseases 2003, 16, (5).
  58. Ciminiello, P.; Fattorusso, E., Bivalve Molluscs as Vectors of Marine Biotoxins Involved in Seafood Poisoning. In Molluscs: From Chemo-ecological Study to Biotechnological Application, Cimino, G.; Gavagnin, M., Eds. Springer Berlin Heidelberg: Berlin, Heidelberg, 2006; pp 53-82.
  59. Gerssen, A.; Klijnstra, M. D., The Determination of Marine Biotoxins in Seafood. In Analysis of Food Toxins and Toxicants, 2017; pp 319-362.
  60. Otero, P.; Silva, M., Emerging Marine Biotoxins in European Waters: Potential Risks and Analytical Challenges. Marine Drugs 2022, 20, (3).
  61. Bakke, M. J.; Horsberg, T. E., Kinetic properties of saxitoxin in Atlantic salmon (Salmo salar) and Atlantic cod (Gadus morhua). Comparative Biochemistry and Physiology Part C: Toxicology & Pharmacology 2010, 152, (4), 444-450.
  62. Kwong, R. W. M.; Wang, W.-X.; Lam, P. K. S.; Yu, P. K. N., The uptake, distribution and elimination of paralytic shellfish toxins in mussels and fish exposed to toxic dinoflagellates. Aquatic Toxicology 2006, 80, (1), 82-91.
  63. Mazzillo, F. F.; Pomeroy, C.; Kuo, J.; Ramondi, P. T.; Prado, R.; Silver, M. W., Angler exposure to domoic acid via consumption of contaminated fishes. Aquatic Biology 2010, 9, (1), 1-12.
  64. Nakamura, M.; Oshima, Y.; Yasumoto, T., Occurrence of saxitoxin in puffer fish. Toxicon 1984, 22, (3), 381-385.
  65. Kershaw, J. L.; Jensen, S.-K.; McConnell, B.; Fraser, S.; Cummings, C.; Lacaze, J.-P.; Hermann, G.; Bresnan, E.; Dean, K. J.; Turner, A. D.; Davidson, K.; Hall, A. J., Toxins from harmful algae in fish from Scottish coastal waters. Harmful Algae 2021, 105, 102068.
  66. Visciano, P.; Schirone, M.; Berti, M.; Milandri, A.; Tofalo, R.; Suzzi, G., Marine Biotoxins: Occurrence, Toxicity, Regulatory Limits and Reference Methods. Frontiers in Microbiology 2016, 7.

Reviewer 3 Report

Comments and Suggestions for Authors

This is a comprehensive and thorough review of fatty acid compositions in marine animals and their possible use in industries. The review is well written for beginners but little information is related with drugs.

Minor comments:

1. Definition of PUFA is ambiguous because the authors write "omega-3 polyunsaturated fatty acids (PUFAs)". Major omega-3 fatty acids are ALA, DHA, and EPA, but ALA is ignored in the manuscript. These are all PUFAs. In all tables, what is the difference between PUFAs and Omega-3? Can we assume PUFA minus Omega-3 equals Omega-6? Please describe how these percentages were calculated from the original articles cited.

2. The major problem of PUFAs is their rapid oxidization and smells. Such disadvantages are not described but should be added in the introductory part.

3. Some parts describe astaxanthin (carotenoid) and steroids, whose structures are different. Probably the authors should describe which category of lipids are discussed early in the review.

Author Response

This is a comprehensive and thorough review of fatty acid compositions in marine animals and their possible use in industries. The review is well written for beginners but little information is related with drugs.

Reply: We thank the reviewer for the comments and suggestions that will allow us to improve our manuscript. In this work, “drugs” are considered in the broader sense, including lipid species and lipophilic compounds that may display bioactivities and even garner pharmacological and cosmeceutical interest. Moreover, this work was submitted to the “Marine Lipids 2023” Special Issue, which also helps substantiate and contextualize the submission of the paper to Marine Drugs.

Minor comments:

  1. Definition of PUFA is ambiguous because the authors write "omega-3 polyunsaturated fatty acids (PUFAs)". Major omega-3 fatty acids are ALA, DHA, and EPA, but ALA is ignored in the manuscript. These are all PUFAs. In all tables, what is the difference between PUFAs and Omega-3? Can we assume PUFA minus Omega-3 equals Omega-6? Please describe how these percentages were calculated from the original articles cited.

Reply: The abbreviation “PUFAs” only encompasses the “polyunsaturated fatty acids” part in that case. Since this was the first time the expression “polyunsaturated fatty acids” appeared in the manuscript, we included the abbreviation, but we understand that this may cause some confusion. Therefore, the abbreviation was removed from the abstract and when associated with “omega-3” in that instance, and included only afterwards to avoid confusion. The fact that ALA is not given the same attention as EPA and DHA is justified by the fact that it only occurs in significantly smaller amounts than EPA and DHA (marine animals are consensually considered the major sources of these fatty acids in the human diet and that is also why they were highlighted here) and that in some studies studies its presence is not even described (for instance [1-3] represent instances in Table 1 where ALA was not reported, as opposed to EPA and DHA). In the tables, “PUFAs” values represent the sum of all (percentages) of fatty acids with more than one unsaturation, while “omega-3” are only a fraction of those (those with an unsaturated bond three carbons from the end of the acyl tail). We can make the assumption that PUFAs minus omega-3 equals omega-6 (there are no other types of polyunsaturated fatty acids reported), but that was not how calculations were made. Like PUFAs, omega-3 and omega-6 were calculated as the sum of the n-3 and n-6 fatty acids present, respectively. Most papers cited already included tables displaying the PUFAs, omega-3 and omega-6 percentages, making our work easier. When that was not the case, these were calculated as the sums of the respective fatty acids, as mentioned before, and when instead of percentages the fatty acids were presented in quantitative units (results expressed as g/kg of lipid or g/100 g of wet weight, for instance), percentages of individual fatty acids were calculated from the total amount of fatty acids described (sum of all the fatty acids present).

  1. The major problem of PUFAs is their rapid oxidization and smells. Such disadvantages are not described but should be added in the introductory part.

Reply: A reference to this disadvantage of dealing with matrixes rich in PUFAs is now included in the text: “In fact, PUFAs are readily degraded by lipid oxidation reactions into a myriad of secondary oxidation products of which short-chain saturated and unsaturated carbonyl compounds (including both aldehydes and ketones) are supposedly the ones contributing the most to flavor deterioration and the occurrence of off-flavors evoking fishy, metallic and rancidity sensations [4-6]. Curiously, it was described that off-odors elicited by the oxidation of PUFAs depend on the specific composition of fatty acids, with different proportions of EPA and DHA modulating the sensory profile [7].” We opted to include this information in the “7. Challenges and Future Directions” section since this section already included a mention to the perishable nature of marine animal co-products and to the expectable oxidation of compounds of interest during storage.

  1. Some parts describe astaxanthin (carotenoid) and steroids, whose structures are different. Probably the authors should describe which category of lipids are discussed early in the review.

Reply: This issue was addressed at the end of the introduction to the “3. Marine Animal Co-Products as a Source Of Healthy Lipids” section: “Anyway, it is well established that triglycerides and phospholipids are the main lipid classes present upon analysis of lipid extracts of marine animals [81-87], and that is also very generally the case for their co-products, as will be later detailed. Other lipid classes that were reported to be present in marine animal co-products and that will be appraised in this section include diacylglycerides and sterols, although the presence of lipid-soluble compounds like carotenoids and vitamins, which should also be present in lipid fractions, will also be highlighted and discussed.”

References:

  1. Lee, S.; Koo, M. H.; Han, D. W.; Kim, I. C.; Lee, J. H.; Kim, J. H.; Sultana, R.; Kim, S. Y.; Youn, U. J.; Kim, J. H., Comparison of Fatty Acid Contents and MMP-1 Inhibitory Effects of the Two Antarctic Fish, Notothenia rossii and Champsocephalus gunnari. Molecules 2022, 27, (14).
  2. Hu, Z.; Chin, Y.; Liu, J.; Zhou, J.; Li, G.; Hu, L.; Hu, Y., Optimization of fish oil extraction from Lophius litulon liver and fatty acid composition analysis. Fisheries and Aquatic Sciences 2022, 25, (2), 76-89.
  3. Khoddami, A.; Ariffin, A.; Bakar, J.; Mohd Ghazali, H., Fatty acid profile of the oil extracted from fish waste (head, intestine and liver) (Sardinella lemuru). World Applied Science Journal 2009, 7.
  4. de Oliveira, D. A. S. B.; Minozzo, M. G.; Licodiedoff, S.; Waszczynskyj, N., Physicochemical and sensory characterization of refined and deodorized tuna (Thunnus albacares) by-product oil obtained by enzymatic hydrolysis. Food Chemistry 2016, 207, 187-194.
  5. Miyashita, K.; Uemura, M.; Hosokawa, M., Effective Prevention of Oxidative Deterioration of Fish Oil: Focus on Flavor Deterioration. Annual Review of Food Science and Technology 2018, 9, (1), 209-226.
  6. Song, G.; Li, L.; Wang, H.; Zhang, M.; Yu, X.; Wang, J.; Shen, Q., Electric Soldering Iron Ionization Mass Spectrometry Based Lipidomics for in Situ Monitoring Fish Oil Oxidation Characteristics during Storage. Journal of Agricultural and Food Chemistry 2020, 68, (7), 2240-2248.
  7. Wen, Y.-Q.; Xue, C.-H.; Zhang, H.-W.; Xu, L.-L.; Wang, X.-H.; Bi, S.-J.; Xue, Q.-Q.; Xue, Y.; Li, Z.-J.; Velasco, J.; Jiang, X.-M., Concomitant oxidation of fatty acids other than DHA and EPA plays a role in the characteristic off-odor of fish oil. Food Chemistry 2023, 404, 134724.

Round 2

Reviewer 2 Report

Comments and Suggestions for Authors

The authors have sufficiently addressed the key concerns of each reviewer. I recommend that this manuscript be considered for publication in current form.